# ON DESIGNING EFFECTIVE RL REWARD AT TRAINING TIME FOR LLM REASONING

## ABSTRACT

Reward models have been increasingly critical for improving the reasoning capability of LLMs. Existing research has shown that a well-trained reward model can substantially improve model performances *at inference time* via search or best-of-N votes. However, the potential of reward models during *RL training time* still remains largely under-explored. It is currently unclear whether these reward models can provide additional training signals to RL training that uses sparse success rewards, which verify the correctness of solutions. In this work, we evaluate popular reward models for RL training, including the Outcome-supervised Reward Model (ORM) and the Process-supervised Reward Model (PRM), and train a collection of LLMs for math problems using RL by combining these learned rewards with success rewards. Surprisingly, even though these learned reward models have strong inference-time performances, they may only bring marginal improvements or even hurt RL *training*, producing worse performances than LLMs trained with the success reward only. We find that *training collapse* easily occurs in RL training when PRM simply serves as reward shaping in addition to the success rewards. Our further analysis reveals two issues that may lead to the sub-optimal performance. Therefore, we introduce two novel reward refinement techniques, including the ***Clip*** and the ***Delta*** mechanisms, to tackle the identified issues. We evaluate our techniques with multiple reward models over a set of 1.5B and 7B LLMs on MATH and GSM8K benchmarks, where both ***Clip*** and ***Delta*** consistently enhance RL training. Finally, we also demonstrate that with a carefully designed reward function, pure RL training without any additional supervised tuning can further improve all the evaluated LLMs, including the state-of-the-art 7B LLM Qwen2.5-Math-7B-Instruct on MATH and GSM8K benchmarks.

## 1 INTRODUCTION

There is a recent trend to improve the reasoning ability of LLMs with learned reward models (Lightman et al., 2024; Wang et al., 2024b; Yu et al., 2024a; Zhang et al., 2024; Lee et al., 2024; Yang et al., 2024b; Luo et al., 2024; Chen et al., 2024c; Havrilla et al., 2024; Shao et al., 2024; Uesato et al., 2022). Recent research has been focusing on guiding search processes during inference (Lightman et al., 2024; Snell et al., 2024; Wang et al., 2024b), with two main categories of reward models: Outcome-supervised Reward Model (ORM) (Cobbe et al., 2021b; Yu et al., 2024a) and Process-supervised Reward Model (PRM) (Lightman et al., 2024; Wang et al., 2024b; Luo et al., 2024). ORM generates *outcome rewards* that estimate the *success rewards*, which evaluate the correctness of generated answers, enabling the selection of the most reliable answer from a pool of generated candidates. By contrast, PRM is trained to distinguish correct reasoning steps from incorrect ones and can provide step-level *process rewards* for search algorithms like Monte-Carlo Tree Search (Chen et al., 2024a) and beam search (Snell et al., 2024).

However, the potential of reward models in RL training for LLM reasoning is not yet fully explored. The most straightforward method for RL training in reasoning tasks is to optimize the success rewards. Some prior works further try the integration of a reward model into RL training (Havrilla et al., 2024; Wang et al., 2024b; Shao et al., 2024). Havrilla et al. (2024) finds that PPO training with a reward model only results in performance degeneration. In addition, some powerful LLMs that exhibit strong reasoning abilities such as the Qwen2.5-Math family (Yang et al., 2024b) and DeepseekMath-7B-RL (Shao et al., 2024) adopt RL training with reward models as a part of their

overall training process for mathematical reasoning. However, due to a lack of detailed analysis on the reward models, it remains unclear whether the reward models can provide additional training signals beyond what the success rewards offer for LLM reasoning.

In this work, we evaluate popular reward models, including ORM and PRM, as RL rewards on the challenging mathematical reasoning benchmark MATH (Hendrycks et al., 2021) and GSM8K (Cobbe et al., 2021a) by using PPO as the RL algorithm (Schulman et al., 2017). Surprisingly, we find that these reward models may only bring marginal benefits to RL training or even lead to performance degradation, yielding even worse results than LLMs trained with a sparse success reward only. We observe that outcome rewards consistently achieve similar training results as success rewards. We hypothesize that outcome rewards may not provide additional training signals since a more accurate success reward is accessible at *training* time. For PRM, we perform an in-depth analysis of the RL training process and observe that RL training easily collapses when simply using the PRM as reward shaping in addition to the success rewards. Through further case studies, we identify two possible causes, including the intrinsic biases of the PRM and a reward misspecification issue, which can be largely exploited by the training LLM to generate sub-optimal behavior patterns.

To tackle these challenges, we propose two novel techniques, i.e., ***Clip*** and ***Delta***, which refine the process rewards for effective RL training. In particular, the ***Clip*** mechanism mitigates the intrinsic biases of PRM. The ***Delta*** mechanism tackles the reward misspecification issue by optimizing single-step PRM rewards. Evaluation of these two techniques on synthetic reasoning trajectories demonstrates that they can effectively mitigate the tendency of training to favor the observed sub-optimal patterns. Finally, we conduct full RL training on a set of advanced 1.5B and 7B LLMs from the Qwen2 and Qwen2.5 families (Yang et al., 2024a;b) with different reward models. Our experiment results show that our proposed techniques effectively enhance RL training. Moreover, with a carefully crafted reward, RL training can improve all the evaluated LLMs, including the state-of-the-art 7B LLM Qwen2.5-Math-7B-Instruct on the challenging MATH and GSM8K (Hendrycks et al., 2021; Cobbe et al., 2021a) benchmarks.

## 2 RELATED WORK

**Reinforcement Learning for LLMs.** In RLHF, Reinforcement learning algorithms can effectively fine-tune LLMs to align with the preference of humans (Dong et al., 2023; Rafailov et al., 2024; Ouyang et al., 2022; Xu et al., 2024; Schulman et al., 2017), to improve the reasoning ability (Shao et al., 2024; Yang et al., 2024b) and coding skills (Wang et al., 2024a; Guo et al., 2024). PPO is the most widely used among the popular RL algorithms due to its robust performance across various domains (Ouyang et al., 2022; Xu et al., 2024). Xu et al. (2024) investigates the implementation details of PPO for dialogue tasks and coding tasks, revealing batch size as a critical factor for improving PPO performance in reinforcement learning from human feedback (RLHF). Our work addresses the challenge of designing RL rewards for LLM reasoning.

**Reward Learning for LLMs.** Learned reward models are widely adopted in RLHF to align LLMs with human preferences (Dong et al., 2023; Rafailov et al., 2024; Ouyang et al., 2022). In RLHF, reward models are trained on binary preference datasets collected from human annotators, following the Bradley-Terry model (Bradley & Terry, 1952). In reasoning tasks involving reliable solution checkers, two main approaches are the Outcome-supervised Reward Model (ORM) (Cobbe et al., 2021b; Yu et al., 2024a) and the Process-supervised Reward Model (PRM) (Lightman et al., 2024; Wang et al., 2024b; Luo et al., 2024). An ORM predicts the correctness of the final answer. A PRM estimates whether the steps so far are correct. Despite the successful applications of reward models, *reward hacking* is a broadly observed issue in learned reward models (Skalse et al., 2022; Singhal et al., 2023; Casper et al., 2023). Through RL training, the LLM may learn to generate high-reward outputs that could not fulfill the intended objectives. Several approaches have been proposed to tackle the reward hacking issue, including disentangling the length aspect of reward modeling (Chen et al., 2024b; Shen et al., 2023), reward ensemble (Eisenstein et al., 2024; Rame et al., 2024), length penalty (Singhal et al., 2023), length normalization (Meng et al., 2024), and various PPO implementation tricks (Singhal et al., 2023; Zheng et al., 2023). In this work, we investigate how to effectively use PRM/ORM as rewards in RL training for LLM reasoning, and

our proposed techniques are related to reward shaping methods in standard RL. (Ng et al., 1999; Harutyunyan et al., 2019; Arjona-Medina et al., 2019; Patil et al., 2020; Widrich et al., 2021) .

**Improving Reasoning Ability of LLMs.** To improve the reasoning ability of LLMs, prior works have focused on several different aspects, including pre-training (Yang et al., 2024b; Achiam et al., 2023; Anil et al., 2023), prompting (Han et al., 2024; Yuan et al., 2024; Wu et al., 2024), search during inference-time (Lightman et al., 2024; Wang et al., 2024b; Yu et al., 2024a; Zhang et al., 2024; Yang et al., 2024b; Luo et al., 2024; Chen et al., 2024c), and fine-tuning (Wang et al., 2024b; Shao et al., 2024; Yang et al., 2024b; Shah et al., 2024; Tang et al., 2024; Yu et al., 2024b). Pre-training methods focus on enriching the data distribution to cover a large amount of rationals and pre-training the LLM over the dataset. The prompting methods elicit the reasoning ability of LLMs through dedicated prompting strategies and automatic agent frameworks. Inference-time search utilizes learned reward models to guide the selection of promising solutions. PRM and ORM could be combined with different search strategies such as Best-of-N, Monte-Carlo Tree Search (Chen et al., 2024a), and Beam Search (Snell et al., 2024). Finally, fine-tuning methods include training the LLM on high-quality question-answer data (Yu et al., 2024b; Shah et al., 2024; Yue et al., 2024) and optimizing the reasoning ability with reinforcement learning (Yang et al., 2024b; Shao et al., 2024; Wang et al., 2024b). In this work, we study how to effectively combine dense and sparse rewards in RL training for reasoning tasks.

## 3 PRELIMINARY

**Language Model.** An LLM is represented as a policy $\pi_\theta(s|q)$ parameterized by $\theta$. In reasoning tasks, $\pi_\theta$ generates a solution $s$ given a question $q$. In addition to the question, $q$ usually also contains a prompt to elicit chain-of-thought reasoning. The solution $s$ is structured with a list of reasoning steps and thus can be viewed from two perspectives, including tokens and steps. From the perspective of tokens, $s$ consists of $T$ tokens, $s = (s_1, s_2, \cdots, s_T)$. From the perspective of steps, $s$ consists of $K$ reasoning steps, $s = (s^{(1)}, s^{(2)}, \cdots, s^{(K)})$ where $s^{(k)}$ denotes the $k$-th reasoning step. For convenience, we use $p^{(k)} = (s^{(1)}, s^{(2)}, \cdots, s^{(k)})$ to denote the solution prefix up to the $k$-th step. In practice, reasoning steps can be parsed with rule-based detectors, enforcing strict output formats, or special tokens (Chen et al., 2024a; Wang et al., 2024b; Lightman et al., 2024).

**Reward Modeling.** In RLHF, the reward models are usually trained with binary preferences (Bradley & Terry, 1952). In reasoning tasks where the correctness of solutions is accessible, reward models can be trained under the supervision of such ground-truth correctness. In reasoning tasks, two primary methods for reward modeling are the Process-supervised Reward Model (PRM) and the Outcome-supervised Reward Model(ORM).

Given a question $q$ and a prefix $s_{1:t}$, an ORM estimates the likelihood the prefix would lead to a correct answer. A standard approach to train an ORM is by first sampling solutions for questions from a dataset with an LLM and then labeling the correctness of each solution. The ORM $r_{\text{outcome}}$ is then trained with the following objective,

$$\mathcal{L}_{\text{ORM}} = \mathbb{E}_{q,s \sim \mathcal{D}} \left[ \sum_{t=1}^{T} \text{Loss}(\text{Correct}(q, s), r_{\text{outcome}}(q, s_{1:t})) \right]$$

where $\text{Correct}(q, s)$ is a binary value indicating the correctness of solution $s$, $t$ enumerates each token of the solution $s$, and Loss denotes the loss function. In practice, the loss function could be binary cross-entropy loss or square-error loss, and we can choose to train ORM on the full sequence or only the last token.

In contrast, Process-supervised Reward Model (PRM) estimates the correctness of individual reasoning steps. PRM is trained with the following objective,

$$\mathcal{L}_{\text{PRM}} = \mathbb{E}_{q,p^{(k)},y_k \sim \mathcal{D}} \left[ \text{Loss}(y_k, r_{\text{process}}(q, p^{(k)})) \right]$$

where $y_k$ is the label for the partial solution $p^{(k)}$ and Loss is the loss function. In practice, binary cross entropy loss is usually adopted. Prior works have investigated several ways to annotate the process labels, including human annotators (Lightman et al., 2024) and automatic annotation with LLMs (Wang et al., 2024b; Luo et al., 2024).

**Reinforcement Learning for LLM Reasoning.** We assume access to the correctness of a solution during training. We use Correct$(q, s)$ to indicate the correctness of solution $s$ to question $q$, which is also referred to as the *success reward* for RL training. An LLM can be fine-tuned to optimize the success reward by using Reinforcement Learning with Kullback-Leibler divergence,

$$J_r(\pi_\theta) = \mathbb{E}_{q \sim \mathcal{D}, s \sim \pi_\theta} \left[ \text{Correct}(q, s) - \beta \log \frac{\pi_\theta(s|q)}{\pi_{ref}(s|q)} \right] \qquad (1)$$

where $\pi_{ref}$ is the reference model for regularizing $\pi_\theta$. Optimizing the success reward only provides a sparse training signal because the reward is provided at the end of the sequence. Alternatively, we can also combine dense rewards with the success reward. The RL objective with dense rewards becomes,

$$J_r(\pi_\theta) = \mathbb{E}_{q \sim \mathcal{D}, s \sim \pi_\theta} \left[ \alpha \cdot \sum_{t=1}^{|s|} r(q, s_{1:t}) + \text{Correct}(q, s) - \beta \log \frac{\pi_\theta(s|q)}{\pi_{ref}(s|q)} \right] \qquad (2)$$

where $r$ denotes the dense reward and $\alpha$ is a coefficient for the dense reward. For example, a PRM $r_{\text{process}}$ can provide dense feedback at the end of reasoning steps, formally represented as $r(q, p^{(k)}) = r_{\text{process}}(q, p^{(k)})$ for any partial solution $p^{(k)}$. In the subsequent sections, we would refer to the rewards generated by ORM as *outcome rewards*, and the rewards generated by PRM as *PRM rewards* to avoid ambiguity between *process rewards* and *dense rewards*.

## 4 RL Reward for LLM Reasoning

In this section, we conduct a systematic study on reward design to aid LLM in learning better reasoning skills through RL training. We follow the RL objective with dense rewards in Eq. (2) and specifically focus on the effective design of dense rewards. As discussed in Sec. 3, the ground-truth correctness, Correct$(p, s)$, serves to provide the sparse rewards, and the dense rewards could be provided by a reward model.

### 4.1 Evaluating RL Training with Learned Reward Models

We first consider two straightforward approaches to apply ORM and PRM to provide rewards in addition to success rewards for RL training. Formally, we consider the following rewards,

- **Solution-Level Outcome Reward (OR):** In the RL training process of Yang et al. (2024b), an ORM provides an estimation of correctness as reward shaping. Note that this is not the case for dense rewards since ORM only produces rewards at the end of the sequence. For a question $q$ and a solution $s$,

$$r(q, s) = r_{\text{outcome}}(q, s) \qquad (3)$$

- **Step-Level Process Reward (PR):** A PRM can provide step-level feedback for RL training. For any solution prefix $p^{(k)}$, dense rewards are the rewards outputted by a PRM,

$$r(q, p^{(k)}) = r_{\text{process}}(q, p^{(k)}) \qquad (4)$$

**Experiment Setup.** We carry out our study on the challenging mathematical reasoning benchmark, MATH (Hendrycks et al., 2021). We use PPO as the RL algorithm and Qwen2-1.5B-Instruct (Yang et al., 2024a) as the base model. For ORM, we sample solutions with the base model and train ORM with binary cross-entropy loss. For PRM, we follow Wang et al. (2024b) to generate process labels with automatic annotation[1]. The ORM and PRM both use Qwen2-1.5B-Instruct as the base model.

---

[1]Implementation details can be found in Sec. 5

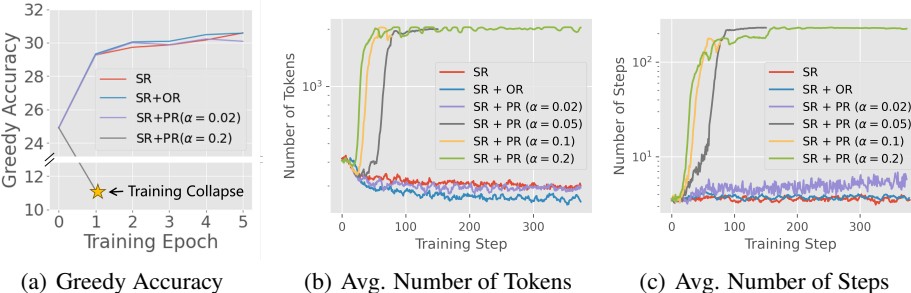

(a) Greedy Accuracy      (b) Avg. Number of Tokens      (c) Avg. Number of Steps

Figure 1: (a) Test greedy accuracy during RL training with a combination of success reward and OR/PR on Qwen2-1.5B-Instruct. SR denotes the success reward. $\alpha$ denotes the reward shaping coefficient. None of OR/PR can surpass training with success rewards. (b) Generation length during RL training. (c) Step count during RL training.

**Results.** Surprisingly, we find these reward functions may not benefit RL training, yielding even worse inference-time performances than LLMs trained with a sparse success reward only, as shown in Fig. 1(a). To further investigate the cause of performance degradation, Fig. 1(b) reports the change in the generation length and the number of reasoning steps during training. Although introducing OR improves the sample efficiency, SR+OR shows a similar evaluation accuracy to adopting a sparse success reward only (SR). We hypothesize this is because an outcome reward may not be able to provide additional information beyond the success reward during training time. On the other hand, when using PRM for RL training, training easily collapses when $\alpha$ is large and we observe a significant increment in the generation length and the number of reasoning steps during RL training. When $\alpha$ is small, SR+PR only achieves sub-optimal performance compared with SR.

**Question**

What is the 10-th term in 1,3,9,15,25,35, ... ?

Step 1: Understand pattern. 0.98
Step 2: Find known numbers. 0.97
Step 3: Establish formula. 0.96
Step 4: Plug numbers into formula. 0.94
Step 5: Solve. 0.93
...
Step 😊. 0.20
Step ready. 0.12
Step nothing. 0.13
<EOS>

**PPO w. Success Reward + PR**
**Return=146.42**

Figure 2: Case study of PR. PRM provides rewards at the end of each step. For PR with a large $\alpha$, the LLM learns to generate many reasoning steps that do not contribute to problem-solving to achieve a high return through RL training.

**Case Study for PR.** For PR with a large $\alpha$ ($\alpha \geq 0.05$), a case study of the generated samples reveals that the LLM learns to obtain high rewards with some specific patterns without faithfully optimizing the ground-truth correctness through RL training. In the generated solutions of PR, there are many short reasoning steps, but these steps only contain unnecessary or meaningless information that does not contribute to problem-solving. As the generation length increases, the model outputs only a single word or even emoji.

**Analysis for PR.** The rewards of unnecessary reasoning steps are positive and could even be large, as shown in the case study (Fig. 2). The LLM learns to exploit this phenomenon by generating more reasoning steps, resulting in a higher return. We further confirm the behavior through some synthetic reasoning trajectories (Fig. 4(a) and Fig. 4(b)), where PR demonstrates extremely larger returns. We identify two key observations through further case studies,

**Observation 1: PRM has *intrinsic biases* to output positive and even high rewards to sub-optimal reasoning steps.** In the left part of Fig. 3, we illustrate a case study of biases of PRM. The PRM could assign high values to steps with simple patterns, such as unnecessary and repetitive steps, even achieving higher rewards than the optimal correct steps. On the other hand, incorrect and nonsense steps could also receive small positive rewards. Consequently RL training would encourage the LLM to generate more sub-optimal steps.

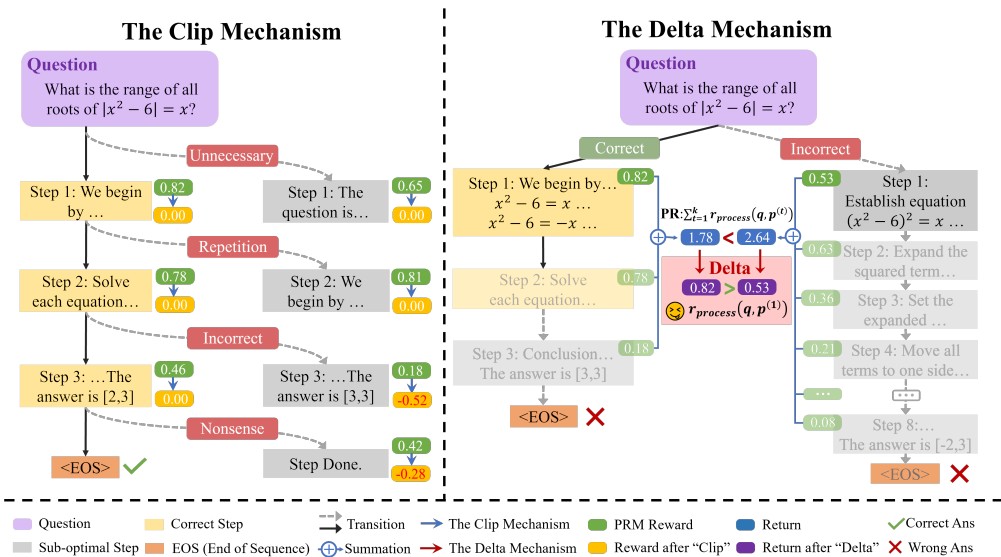

Figure 3: **Left: A case study of the intrinsic biases of PRM & the effect of the Clip mechanism.** PRM may assign high values to steps with simple patterns, such as unnecessary steps and repetitive ones, even higher than the correct steps. For incorrect and nonsense steps, PRM may assign small positive values. *The Clip mechanism can mitigate the intrinsic biases of PRM, preventing the LLM from obtaining high rewards through these undesired patterns and penalizing steps that have low PRM rewards.* **Right: A reward misspecification case of PR showing that RL training promotes an incorrect step & the effect of the Delta mechanism.** In this case, the left solution starts with two correct steps but has an incorrect answer. The right solution starts with an incorrect step and also has an incorrect answer. Although the first correct step in the left solution has a higher PRM reward than the first incorrect step in the right solution, RL training with PR would prefer the incorrect step. This is because the incorrect step receives a higher return than the correct step since the right solution accumulates PRM rewards of much more steps. *The Delta mechanism ensures the steps promoted by RL training are aligned with the PRM, which promotes the correct step in this case.*

**Observation 2: RL training may mistakenly prefer an incorrect step with a low PRM reward, as shown by the case study in the right part of Fig. 3.** Although the PRM successfully assigns a low PRM reward to the incorrect reasoning step, RL training would mistakenly encourage the incorrect step. We refer to this issue as the *reward misspecification* (Pan et al., 2022) issue since the RL objective of PR does not align with the desired target, i.e. better reasoning skills, even when the PRM can accurately assign higher rewards to better reasoning steps. Furthermore, We find such an issue also exists in PRMs that are trained with human-labeled data. (See Appendix F)

Here are two key takeaways regarding applying ORM and PRM in RL training,

**Takeaway for ORM.** For ORM, it only marginally improves the sample efficiency of RL training compared with the sparse success reward. We hypothesize this is because, when a success reward is available during training time, ORM does not provide additional supervision signal and should not be a preferred choice at RL training time. We also remark that ORM does not suffer from training collapse since OR only uses ORM to generate sparse rewards, and the ORM outputs a 0-1 value, naturally ensuring an upper-bounded objective.

**Takeaway for PRM.** Simply adopting PRM rewards as dense rewards would easily lead to training collapse during RL training. Although PRM provides useful training signals for intermediate steps, issues including the intrinsic biases of the learned PRM and the reward misspecification issue could cause the LLM to generate sub-optimal behavior patterns through RL training.

## 4.2 CONSTRUCTING EFFECTIVE RL REWARDS WITH PRM

Since ORM does not provide dense feedback for RL training and may lack additional information beyond the success reward during training, PRM can be a more suitable source for dense rewards.

However, as analyzed in Sec. 4.1, the intrinsic biases of PRM and the issue of reward misspecification pose challenges for adopting PRM to promote better reasoning skills in RL training. To effectively unleash the potential of PRM in RL training, we introduce two novel techniques designed to utilize PRM in RL training effectively,

**The *Clip* mechanism.** We propose the *Clip* mechanism to mitigate the intrinsic biases of PRM. To prevent the LLM from exploiting the reward model to achieve high rewards with undesired patterns such as repetition and unnecessary reasoning, a straightforward idea is to upper-bound high rewards by a selected threshold $\eta$. For steps with low PRM rewards, we penalize such steps with a negative reward $r_{\text{process}}(q, p^{(k)}) - \eta$. Formally, with a threshold $\eta$,

$$r(q, p^{(k)}) = \min(r_{\text{process}}(q, p^{(k)}) - \eta, 0) \tag{5}$$

If a suitable $\eta$ is chosen, the majority of the reasoning steps would receive a reward of 0, and only steps with low $r_{\text{process}}$ would have a negative reward. In practice, for each question in a training batch, by default we set $\eta$ to be the average PRM rewards of all reasoning steps.

**The *Delta* mechanism.** We also introduce the *Delta* mechanism to tackle reward misspecification issue. The *Delta* mechanism subtracts the rewards between adjacent steps. Specially, the reward for the last reasoning step is dropped since the success reward would be sufficient to provide guidance for the last reasoning step. Formally, for a solution prefix $p^{(k)}$,

$$r(q, p^{(k)}) = \begin{cases} r_{\text{process}}(q, p^{(k)}) - r_{\text{process}}(q, p^{(k+1)}) & \text{if } k < K - 1 \\ r_{\text{process}}(q, p^{(k)}) & \text{if } k = K - 1 \\ 0 & \text{if } k = K \end{cases} \tag{6}$$

A nice property of the Delta mechanism is that it ensures the return starting from any intermediate solution step $p^{(k)}$ is $\alpha \cdot r_{\text{process}}(q, p^{(k)}) + \text{Correct}(q, s)$, which is unaffected by the PRM rewards of future steps. Following (Sutton, 2018), the policy gradient of combining the PRM rewards processed with the Delta mechanism and sparse success rewards is (informally) given by,

$$\nabla_\theta J_r(\pi_\theta) = \mathbb{E}_{q \sim \mathcal{D}, s \sim \pi_\theta(\cdot|q)} [\nabla_\theta \log \pi_\theta(s|q) \cdot \text{Correct}(q, s) \tag{7}$$

$$+ \alpha \cdot \underbrace{\sum_{k=1}^{K-1} \nabla_\theta \log \pi_\theta(s^{(k)}|q, p^{(k-1)}) \cdot r_{process}(q, p^{(k)})}_{\text{Effect of the Delta mechanism}}] + \text{KL term}$$

$$\tag{8}$$

Consequently, RL training would focus on optimizing single-step PRM rewards. The theoretical analysis can be found in Appendix. E.

These mechanisms can be used individually or in combination. In practice, we consider three approaches incorporating these mechanisms: (1) *PR-Clip*, which applies the Clip mechanism on the PRM rewards, (2) *PR-Delta*, which applies the Delta mechanism, and (3) *PR-Clip-Delta*, which first applies the Clip mechanism and then the Delta mechanism.

We further perform evaluation on synthetic solutions that exhibit repetitive patterns in different ways. As shown in Fig. 4(b) and Fig. 4(a), the Clip mechanism and the Delta mechanism can both successfully limit the upper bound of the returns on these synthetic solutions. Additionally, the Clip mechanism imposes increasingly smaller returns as the length of the repetitive pattern grows.

**Other Practices.** We also compare with some adopted practices to avoid reward hacking in prior works (Singhal et al., 2023), including length normalization and length penalty. More details can be found in Appendix C. Length normalization normalizes the rewards for each solution. Length penalty imposes a constant penalty for each step. As illustrated in Fig. 4, imposing length penalty and length normalization could still favor the undesired repetition modes over correct solutions. We also investigate standard normalization for PRM as employed by Shao et al. (2024), which we find would lead to training instability. More details can be found in Sec. 5.2.

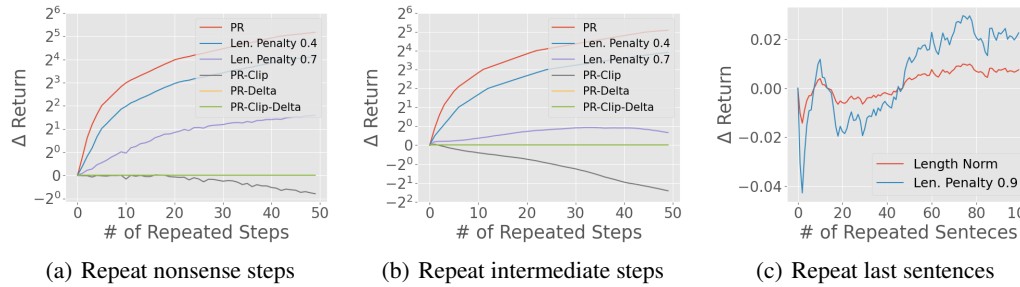

(a) Repeat nonsense steps     (b) Repeat intermediate steps     (c) Repeat last sentences

Figure 4: Difference between the returns of synthetic solutions and the ground-truth solution. The synthetic solutions are constructed from the ground-truth solution by (a) repeatedly adding nonsense steps to the end of the sequence, (b) repeating an intermediate step, and (c) repeating the last sentence in the solution. A positive return difference indicates the repetitive patterns are favored over the ground-truth solution. Both PR and length penalty can gain significantly high returns in (a) and (b). In (c), when sufficient repetitions are inserted, length normalization and length penalty would assign a higher return to the synthetic solution than to the ground-truth solution.

## 5 EXPERIMENTS

### 5.1 EXPERIMENT SETUP

**Training Dataset.** We conduct RL training on the MathInstruct (Yue et al., 2024) dataset. In particular, we only use the questions and the golden answers in the dataset while the provided solutions are not used for training. We use Qwen2-7B-Instruct to sample 16 answers for each question in the training dataset and keep those questions that have both correct and wrong answers. To train an ORM, binary cross entropy loss is adopted. For PRM training, we follow Wang et al. (2024b) to generate automatic process annotations by using Qwen2-7B-Instruct as the completer. Specifically, for each step in the generated samples, we sample 8 solutions starting from the solution prefix. This step is labeled as correct if any of these 8 solutions is correct.

**Benchmarks & Metrics.** We carry out our evaluation on the GSM8K (Cobbe et al., 2021a) and MATH (Hendrycks et al., 2021) datasets. For evaluation metrics, we report the *Greedy* and *Sampling* scores, which correspond to adopting greedy decoding and sampling with temperature of 1 as generation strategies, respectively.

**Base Models.** Our experiments are taken over a series of large language models from the Qwen2 (Yang et al., 2024a) family and the state-of-the-art LLMs for mathematical reasoning, Qwen2.5 (Yang et al., 2024b) family. Specifically, we use various 1.5B and 7B LLMs, including general and math-specific models. For general models, we consider Qwen2-1.5B-Instruct and Qwen2-7B-Instruct. For math-specific models, we consider Qwen2-Math-1.5B-Instruct, Qwen2.5-Math-1.5B-Instruct, Qwen2-Math-7B-Instruct and Qwen2.5-Math-7B-Instruct. Note that these LLMs already equip sufficient instruction following ability and we do not perform any further supervised fine-tuning. Lastly, the PRM is trained with the same base model as the actor model.

**RL Training** We adopt the Proximal Policy Optimization (PPO) implementation of ReaLHF (Mei et al., 2024), which supports LLM fine-tuning with dense rewards. The detailed hyperparameters and training setup are listed in Appendix. D.

### 5.2 ABLATION STUDY

**The Clip & Delta Mechanisms** Our ablation study of the Clip and Delta mechanisms is presented in Table 1. We also consider a standard normalization variant of PR (Shao et al., 2024), denoted as PR-Normed. PPO training with OR can not surpass training with a sparse success reward. PR only achieves sub-optimal performance. Similarly, the performance of PR-Normed also decreases in the latter epochs. Consequently, none of OR, PR, and PR-Normed can achieve higher greedy decoding accuracy than training with a success reward. On the other hand, the Delta mechanism successfully stabilizes RL training, surpassing training with a success reward. Finally, by combining

| Method | Greedy | Sampling |
|---|---|---|
| Qwen2-1.5B-Instruct | 24.90 | 16.79 |
| Success Reward | 30.58 | 27.05 |
| SR + OR | 30.57 | 27.12 |
| SR + PR (**E4**) | 30.22 | 27.46 |
| SR + PR-Normed (**E2**) | 29.66 | 27.14 |
| SR + PR-Normed (**E5**) | 12.36 | 12.84 |
| SR + PR-Clip | 30.30 | **28.40** |
| SR + PR-Delta | 30.68 | 27.96 |
| SR + PR-Clip-Delta | **31.44** | 28.20 |

Table 1: Ablation study of various reward functions with Qwen2-1.5B-Instruct. **E2** denotes the results of the 2-nd epoch. Unless otherwise specified, we report the accuracy of final checkpoint.

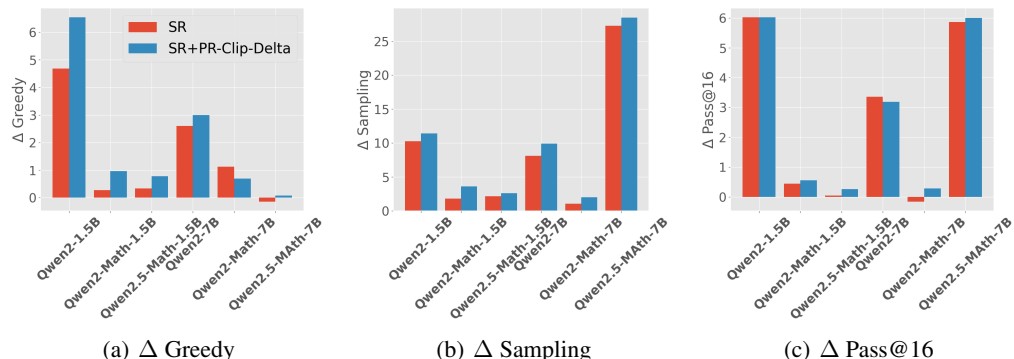

(a) $\Delta$ Greedy      (b) $\Delta$ Sampling      (c) $\Delta$ Pass@16

Figure 5: Performance improvement of PPO training over the base LLMs using success rewards and further using PR-Clip-Delta as dense rewards. All LLMs are the instruction following model, and the "-Instruct" suffices are omitted for improved clarity. Adding PR-Clip-Delta as dense rewards consistently improves RL training with sparse success rewards only.

the Clip mechanism and the Delta mechanism, PR-Clip-Delta demonstrates the best greedy decoding accuracy. Training curves of all approaches are provided in Appendix. A.2

**Effect of PR-Clip-Delta** We compare the performance improvements of PPO training over the base LLMs when using a success reward and additionally using PR-Clip-Delta as dense rewards in Fig. 5. In addition to Greedy and Sampling scores, we also consider the Pass@16 score, which we believe can roughly estimate the upper bound of the model's capacity. Using PR-Clip-Delta as dense rewards can consistently improve RL training, across all LLMs and all evaluation metrics, except the greedy decoding accuracy on Qwen2-Math-7B-Instruct. This suggests that applying the Clip mechanism and the Delta mechanism can effectively utilize the PRM to guide the LLM in learning better reasoning skills during RL training. We report the detailed numbers in Appendix A.

## 5.3 MAIN RESULTS

**Main Results** Our main results are summarized in Table. 2. RL training consistently improves the performance of the base model across all the models we test, even on the state-of-the-art 1.5B model, Qwen2.5-Math-1.5B-Instruct, and 7B model, Qwen2.5-Math-7B-Instruct. For 1.5B models, Qwen2-1.5B-Instruct obtains the most significant performance improvement. Through RL training with PR-Clip-Deta as reward function, the best 1.5B model, Qwen2.5-Math-1.5B-Instruct achieves 87.34% and 76.78% greedy decoding accuracy on GSM8K and MATH benchmark respectively, indicating 2.20% and 0.78% improvement of accuracy over the base model. For 7B models, building on the strongest 7B LLM, Qwen2.5-Math-7B-Instruct, RL training with dense reward further boosts the performance and achieves 95.6% and 83.38% greedy decoding accuracy on GSM8K and MATH benchmarks, respectively, surpassing several baselines. It is noteworthy that Qwen2.5-Math-7B-Instruct is already trained using RL, highlighting the effectiveness of PR-Clip-Delta.

**Performance Improvement** The performance improvement of RL training varies across models with different amounts of parameters and different strengths. In general, weaker models gain higher

| Model | GSM8K | | MATH | |
|---|---|---|---|---|
| | Greedy | Sampling | Greedy | Sampling |
| GPT-4o-2024-08-06 | 92.9 | - | 81.1 | - |
| DeepSeekMath-7B-RL | 88.2 | - | 52.4 | - |
| Internlm2-math-plus-7B | 84.0 | - | 54.4 | - |
| Mathstral-7B-v0.1 | 84.9 | - | 56.6 | - |
| NuminaMath-7B-CoT | 75.4 | - | 55.2 | - |
| Llama-3.1-8B-Instruct | 76.6 | - | 47.2 | - |
| *1.5B Models* | | | | |
| Qwen2-1.5B-Instruct | 50.19 | 44.58 | 24.90 | 16.79 |
| + PPO w. SR | $67.70^{\uparrow 17.51}$ | $65.50^{\uparrow 20.92}$ | $30.58^{\uparrow 4.68}$ | $27.05^{\uparrow 10.26}$ |
| + PPO w. (SR + PR-Clip-Delta) | $\mathbf{68.76}^{\uparrow 18.57}$ | $\mathbf{66.19}^{\uparrow 21.61}$ | $\mathbf{31.44}^{\uparrow 6.54}$ | $\mathbf{28.20}^{\uparrow 11.41}$ |
| Qwen2-Math-1.5B-Instruct | 83.62 | 81.50 | 69.98 | 64.51 |
| + PPO w. SR | $84.61^{\uparrow 0.99}$ | $83.93^{\uparrow 2.43}$ | $70.26^{\uparrow 0.28}$ | $66.29^{\uparrow 1.78}$ |
| + PPO w. (SR + PR-Clip-Delta) | $\mathbf{85.67}^{\uparrow 2.05}$ | $\mathbf{84.76}^{\uparrow 3.26}$ | $\mathbf{70.94}^{\uparrow 0.96}$ | $\mathbf{68.13}^{\uparrow 3.62}$ |
| Qwen2.5-Math-1.5B-Instruct | 85.14 | 82.11 | 76.00 | 72.05 |
| + PPO w. SR | $86.73^{\uparrow 1.59}$ | $85.82^{\uparrow 3.71}$ | $76.34^{\uparrow 0.34}$ | $74.22^{\uparrow 2.17}$ |
| + PPO w. (SR + PR-Clip-Delta) | $\mathbf{87.34}^{\uparrow 2.20}$ | $\mathbf{85.97}^{\uparrow 3.86}$ | $\mathbf{76.78}^{\uparrow 0.78}$ | $\mathbf{74.63}^{\uparrow 2.58}$ |
| *7B Models* | | | | |
| Qwen2-7B-Instruct | 86.88 | 80.44 | 57.54 | 48.27 |
| + PPO w. SR | $87.72^{\uparrow 0.84}$ | $86.81^{\uparrow 6.37}$ | $60.14^{\uparrow 2.60}$ | $56.39^{\uparrow 8.12}$ |
| + PPO w. (SR + PR-Clip-Delta) | $\mathbf{87.64}^{\uparrow 0.76}$ | $\mathbf{87.34}^{\uparrow 6.90}$ | $\mathbf{60.54}^{\uparrow 3.00}$ | $\mathbf{58.17}^{\uparrow 9.90}$ |
| Qwen2-Math-7B-Instruct | 89.61 | 89.23 | 75.30 | 72.09 |
| + PPO w. SR | $89.46^{\downarrow 0.15}$ | $90.07^{\uparrow 0.84}$ | $\mathbf{76.42}^{\uparrow 1.12}$ | $73.12^{\uparrow 1.03}$ |
| + PPO w. (SR + PR-Clip-Delta) | $\mathbf{90.90}^{\uparrow 1.29}$ | $\mathbf{90.14}^{\uparrow 0.91}$ | $76.00^{\uparrow 0.70}$ | $\mathbf{74.09}^{\uparrow 2.00}$ |
| Qwen2.5-Math-7B-Instruct | 95.60 | 80.74 | 83.30 | 52.76 [2] |
| + PPO w. SR | $95.45^{\downarrow 0.15}$ | $\mathbf{95.07}^{\uparrow 14.33}$ | $83.16^{\downarrow 0.14}$ | $79.95^{\uparrow 27.19}$ |
| + PPO w. (SR + PR-Clip-Delta) | $\mathbf{95.60}^{0.00}$ | $\mathbf{95.07}^{\uparrow 14.33}$ | $\mathbf{83.38}^{\uparrow 0.08}$ | $\mathbf{81.22}^{\uparrow 28.46}$ |

Table 2: Greedy and Sampling scores on GSM8K and MATH benchmarks. PPO training using sparse success rewards and PR-Clip-Delta as dense rewards consistently improve all evaluated LLMs, including the state-of-the-art 7B LLMs, Qwen2.5-Math-7B-Instruct. For sampling decoding, we adopt the temperature of 1.0.

performance improvements than stronger models. Comparing the improvements of Greedy and Sampling scores, the improvements of Sampling score are larger than those of Greedy score across all LLMs, resulting in a smaller gap between Sampling and Greedy scores.

# 6 CONCLUSION

In this work, we investigate designing dense rewards with a process-supervised reward model in RL training to improve the reasoning ability of LLMs. We examine some popular reward models and identify the issue of reward hacking, which manifests as the generation of nonsensical texts or unnecessary reasoning steps. The reward hacking issue can be mitigated with our proposed techniques, using the Clip mechanism to prevent the LLM from exploiting the reward model and the Delta mechanism to ensure a bounded RL objective. We show that the proposed techniques can be utilized to apply Process-supervised Reward Models for improved RL training.

**Limitations.** Limited by computation resources, our experiments are conducted over 1.5B&7B LLMs, while evaluations on larger LLMs could further help verify our proposed techniques. Also, it is an interesting direction to perform various inference-time search strategies with the LLMs trained with PPO, which could help further understand whether RL training can improve search performance. Furthermore, we believe that with the support of more powerful reward models, RL training can bring greater benefits to LLM reasoning.

---

[2] For sampling accuracy, we find that Qwen-2.5-math-Instruct is likely to generate strange characters, leading to poor sampling accuracy.

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

# A ADDITIONAL RESULTS

## A.1 MAIN RESULTS

In Tab. 3 and Tab. 4, we report the results of RL training on different base models, including those with success rewards and after applying PR-Clip-Delta.

| Model | Method | Math | | |
|---|---|---|---|---|
| | | Greedy | Sample | Pass@16 |
| Qwen2-1.5B-Instruct | Basemodel | 24.90 | 16.79 | 55.68 |
| | Success Reward | $30.58^{\uparrow 4.68}$ | $27.05^{\uparrow 10.26}$ | $61.70^{\uparrow 6.02}$ |
| | + PR-Clip-Delta | $31.44^{\uparrow 6.54}$ | $28.20^{\uparrow 11.41}$ | $61.70^{\uparrow 6.02}$ |
| Qwen2-Math-1.5B-Instruct | Basemodel | 69.98 | 64.51 | 88.02 |
| | Success Reward | $70.26^{\uparrow 0.28}$ | $66.29^{\uparrow 1.78}$ | $88.46^{\uparrow 0.44}$ |
| | + PR-Clip-Delta | $70.94^{\uparrow 0.96}$ | $68.13^{\uparrow 3.62}$ | $88.58^{\uparrow 0.56}$ |
| Qwen2.5-Math-1.5B-Instruct | Basemodel | 76.00 | 72.05 | 90.50 |
| | Success Reward | $76.34^{\uparrow 0.34}$ | $74.22^{\uparrow 2.17}$ | $90.54^{\uparrow 0.04}$ |
| | + PR-Clip-Delta | $76.78^{\uparrow 0.78}$ | $74.63^{\uparrow 2.58}$ | $90.76^{\uparrow 0.26}$ |
| Qwen2-7B-Instruct | Basemodel | 57.54 | 48.27 | 80.04 |
| | Success Reward | $60.14^{\uparrow 2.60}$ | $56.39^{\uparrow 8.12}$ | $83.40^{\uparrow 3.36}$ |
| | + PR-Clip-Delta | $60.54^{\uparrow 3.00}$ | $58.17^{\uparrow 9.90}$ | $83.22^{\uparrow 3.18}$ |
| Qwen2-Math-7B-Instruct | Basemodel | 75.30 | 72.09 | 91.24 |
| | Success Reward | $76.42^{\uparrow 1.12}$ | $73.12^{\uparrow 1.03}$ | $91.08^{\downarrow 0.16}$ |
| | + PR-Clip-Delta | $76.00^{\uparrow 0.70}$ | $74.09^{\uparrow 2.00}$ | $91.52^{\uparrow 0.28}$ |
| Qwen2.5-Math-7B-Instruct | Basemodel | 83.3 | 52.76 | 86.6 |
| | Success Reward | $83.16^{\downarrow 0.14}$ | $79.95^{\uparrow 27.19}$ | $92.46^{\uparrow 5.86}$ |
| | + PR-Clip-Delta | $83.38^{\uparrow 0.08}$ | $81.22^{\uparrow 28.46}$ | $92.60^{\uparrow 6.00}$ |

Table 3: Results on MATH test set

In Fig. 6, we report the greedy accuracy on MATH test set of different training epochs, where epoch-0 means the base model (i.e., Qwen2-1.5B-Instruct). The introduction of PR-norm caused the model's accuracy to drop significantly starting from the third epoch.

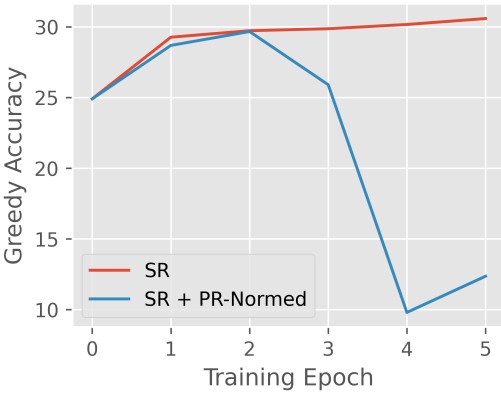

Figure 6: Greedy accuracy on MATH test set during the training process.

| Model | Method | GSM8K | |
|---|---|---|---|
| | | Greedy | Sample |
| Qwen2-1.5B-Instruct | Basemodel | 50.19 | 44.58 |
| | Success Reward
+ PR-Clip-Delta | $67.70^{\uparrow 17.51}$
$68.76^{\uparrow 18.57}$ | $65.50^{\uparrow 20.92}$
$66.19^{\uparrow 21.61}$ |
| Qwen2-Math-1.5B-Instruct | Basemodel | 83.62 | 81.50 |
| | Success Reward
+ PR-Clip-Delta | $84.61^{\uparrow 0.99}$
$85.67^{\uparrow 2.05}$ | $83.93^{\uparrow 2.43}$
$84.76^{\uparrow 3.26}$ |
| Qwen2.5-Math-1.5B-Instruct | Basemodel | 85.14 | 82.11 |
| | Success Reward
+ PR-Clip-Delta | $86.73^{\uparrow 1.59}$
$87.34^{\uparrow 2.20}$ | $85.82^{\uparrow 3.71}$
$85.97^{\uparrow 3.86}$ |
| Qwen2-7B-Instruct | Basemodel | 86.88 | 80.44 |
| | Success Reward
+ PR-Clip-Delta | $87.72^{\uparrow 0.84}$
$87.64^{\uparrow 0.76}$ | $86.81^{\uparrow 6.37}$
$87.34^{\uparrow 6.90}$ |
| Qwen2-Math-7B-Instruct | Basemodel | 89.61 | 89.23 |
| | Success Reward
+ PR-Clip-Delta | $89.46^{\downarrow 0.15}$
$90.90^{\uparrow 1.29}$ | $90.07^{\uparrow 0.84}$
$90.14^{\uparrow 0.91}$ |
| Qwen2.5-Math-7B-Instruct | Basemodel | 95.60 | 80.74 |
| | Success Reward
+ PR-Clip-Delta | $95.45^{\downarrow 0.15}$
$95.60^{\uparrow 0.00}$ | $95.07^{\uparrow 14.33}$
$95.07^{\uparrow 14.33}$ |

Table 4: Results on GSM8K test set

## A.2 TRAINING CURVES

We list training curves of all methods on Qwen2-1.5B-Instruct here.

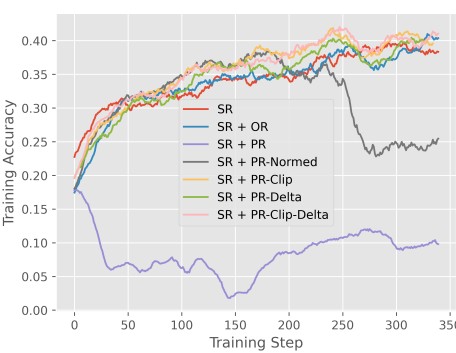

Figure 7: Training accuracy of all baselines and the proposed methods on Qwen2-1.5B-Instruct

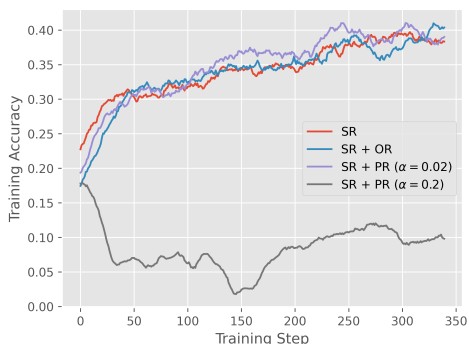

Figure 8: Training accuracy of SR, SR+OR, and SR+PR on Qwen2-1.5B-Instruct

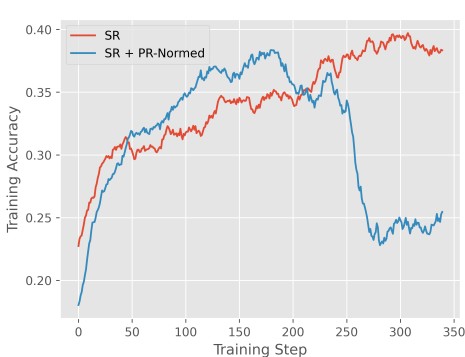

Figure 9: Training accuracy of SR and SR+PR-Normed on Qwen2-1.5B-Instruct

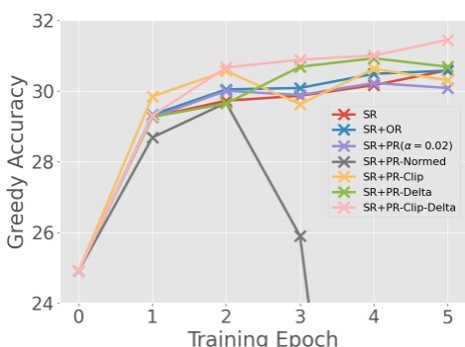

Figure 10: Test accuracy of all methods across training epochs on Qwen2-1.5B-Instruct

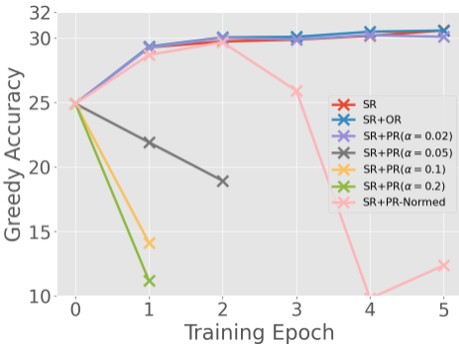

Figure 11: Test accuracy of all baselines across training epochs on Qwen2-1.5B-Instruct

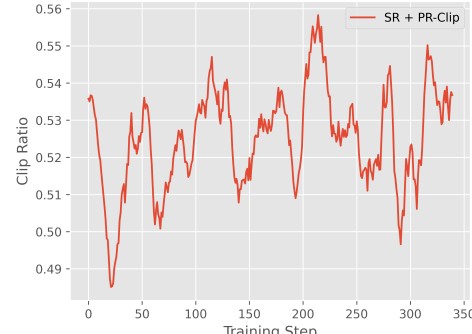

Figure 12: Clip ratio of PR-Clip during RL training. PR-Clip indeed provides dense rewards for RL training.

## A.3 ABLATION STUDIES

We perform ablation study for the reward shaping coefficient $\alpha$ in Tab. 5. The ablation study for threshold $\eta$ in PR-Clip is provided in Fig. 13.

|  | Epoch 1 | Epoch 2 | Epoch 3 | Epoch 4 | Epoch 5 |
|---|---|---|---|---|---|
| SR | 29.26 | 29.72 | 29.86 | 30.16 | 30.58 |
| SR+PR ($\alpha = 0.02$) | 29.25 | 30.00 | 29.88 | 30.22 | 30.08 |
| SR+PR ($\alpha = 0.05$) | 21.90 | 18.92 | / | / | / |
| SR+PR ($\alpha = 0.1$) | 14.10 | / | / | / | / |
| SR+PR ($\alpha = 0.2$) | 11.16 | / | / | / | / |

Table 5: Test accuracy of SR vs. SR+PR with different reward shaping coefficient $\alpha$ across training epochs

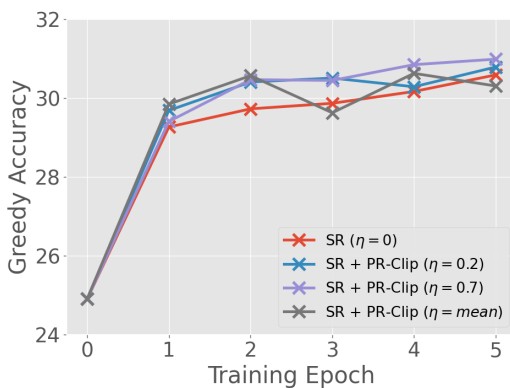

Figure 13: Ablation study of the threshold $\eta$ of PR-Clip on Qwen2-1.5B-Instruct. Greedy accuracy is reported. '$\eta$ =mean' means that we set $\eta$ to be the average PRM rewards of all reasoning steps related to a question in a training batch. A constant $\eta$ offers more stable improvement.

## B  SYNTHETIC EXAMPLE

In Fig. 4, we insert nonsense steps or repeat the last sentences in the following manner.

---

**Prompt:**

Tom found a box of 220 baseball cards. Being a thoughtful older brother, he wanted to save at least 50 percent of the cards for himself and split the remaining cards evenly among his 5 younger brothers. What is the lowest possible number of cards that Tom can save for himself that will also allow each of his 5 younger brothers to get an equal number of cards? Please reason step by step, and put your final answer within \boxed{}.

**Answer by Qwen2-1.5B-instruct:**

Step 1: 50% of 220 cards

$$50\% \times 220 = 0.50 \times 220 = 110$$

Therefore, Tom would save 110 of the cards for himself.

Step 2: Remaining cards that Tom would like to distribute evenly among his 5 younger brothers

$$220 - 110 = 110$$

Tom would have 110 cards left to distribute evenly among his 5 younger brothers.

Step 3: Forming sets of cards and evenly distributing

If the 110 remaining cards are to be divided into 5 sets of equal cards, each set must contain:

---

$$\frac{110}{5} = 22$$

Therefore, each of Tom's 5 younger brothers would get 22 cards, but since each set also must have 22 cards, the lowest possible number of cards for Tom to save so that each brother gets an equal number of cards means Tom would save:

$$5 \times 22 = 110$$

The lowest possible number of cards that Tom can save for himself that will allow each of his 5 younger brothers to get an equal number of cards is 110. This way, Tom will save enough cards to permit a remainder that will also allow one card to be shared by each of his brothers out of the total of 110 cards.

The answer is: $\boxed{110}$ .

**Repeat Nonsense**

Step 1: 50% of 220 cards
$$50\% \times 220 = 0.50 \times 220 = 110$$

...

Step 3: Forming sets of cards and evenly distributing ...

Step done.

Step done.

Step end.

Step end.

Step show.

...

**Repeat Mid Step**

Step 1: 50% of 220 cards
$$50\% \times 220 = 0.50 \times 220 = 110$$

...

Step 2: Remaining cards that Tom would like to distribute evenly among his 5 younger...

Step 2: Remaining cards that Tom would like to distribute evenly among his 5 younger...

Step 2: Remaining cards that Tom would like to distribute evenly among his 5 younger...

...

Step 3: Forming sets of cards and evenly distributing ...

The answer is: $\boxed{110}$ .

**Repeat Last Sentence**

Step 1: 50% of 220 cards
$$50\% \times 220 = 0.50 \times 220 = 110$$

...

Step 3: Forming sets of cards and evenly distributing ...

The answer is: 110 .

The answer is: 110 .

The answer is: 110 .

The answer is: 110 .

...

## C  BASELINES

**Length Normalization.**  Length normalization normalizes the rewards for each solution. Formally,

$$r(q, p^{(k)}) = \frac{1}{K} r_{\text{process}}(q, p^{(k)})$$

**Length Penalty.**  Length penalty imposes a constant penalty for each step.

$$r(q, p^{(k)}) = r_{\text{process}}(q, p^{(k)}) - k * c_{\text{penalty}}$$

## D  HYPER-PARAMETERS AND TRAINING SETUP

**PPO Hyperparameters and Training Setup.**  Following prior practices (Shao et al., 2024; Xu et al., 2024), we adopt a large batch size and sample multiple solutions for each question within a batch. We find a large batch size and multiple sampling critical to the overall performance of RL training. For 1.5B models, there are 1024 questions, and 8 solutions are sampled for each question in a batch, leading to a batch size of $1024 \times 8$. For 7B models, the batch size is $4096 \times 8$. Each training batch is split into 4 minibatches. We apply a KL penalty coefficient of 0.1, a coefficient of 1 for dense rewards, and a coefficient of 5 for successful rewards. For the reward threshold $\eta$ in the Clip mechanism, by default the average value of PRM rewards of all reasoning steps related to one question in a training batch. The learning rates of 1B and 7B actor models are 1e-6 and 1e-5, respectively, while all critic models use a learning rate of 5e-6. We use Adam optimizer weight decay of 0.05. The 1.5B models are trained on a cluster of 4 machines, each with 8 Nvidia H100 GPUs, for approximately 8 hours. The 7B models are trained on a cluster of 8 machines, each with 8 Nvidia H100 GPUs, for approximately 20 hours.

**Implementation of Clip and Delta.**  Both the Clip and the Delta mechanisms are straightforward to integrate into existing workflows. For the Clip mechanism, its implementation involves computing the mean of the reward as a threshold after reward calculation, followed by applying the formula specified in Eq. 5. This additional step is computationally lightweight and seamlessly fits within the existing reward processing pipeline. The Delta mechanism requires computing the difference between rewards from two adjacent steps, a process that is both conceptually simple and computationally efficient. As such, neither method introduces significant overhead, ensuring their ease of adoption.

## E  THEORETICAL ANALYSIS

### E.1  RL TRAINING WITH SR+PR-DELTA

**Lemma 1 (Policy gradient of RL training with SR+PR-Delta)** *Following (Sutton, 2018), the policy gradient of the RL objective combining the PRM rewards processed with the Delta mech-*

*anism and the sparse success reward (SR+PR-Delta) is given by,*

$$\nabla_\theta J_r(\pi_\theta) = \mathbb{E}_{q\sim\mathcal{D},s\sim\pi_\theta(\cdot|q)}[\sum_{k=1}^{K}\nabla_\theta \log \pi_\theta(s^{(k)}|q,p^{(k-1)}) \cdot (\alpha \cdot r_{process}(q,p^{(k)}) \cdot \mathbb{I}[k < K] \quad (9)$$

$$+ Correct(q,s) - \beta \cdot \sum_{k'=k}^{K} \log \frac{\pi_\theta(s^{(k')}|q,p^{(k'-1)})}{\pi_{ref}(s^{(k')}|q,p^{(k'-1)})})] \quad (10)$$

**Proof 1** *Following (Sutton, 2018), the vanilla policy gradient is given by,*

$$\nabla_\theta J_r(\pi_\theta) = \mathbb{E}_{q\sim\mathcal{D},s\sim\pi_\theta(\cdot|q)}[\sum_{k=1}^{K}\nabla_\theta \log \pi_\theta(s^{(k)}|q,p^{(k-1)})G_k] \quad (11)$$

*where $G_k$ is the return from the $k$-th reasoning step.*
*By the formula of the Delta mechanism (Eq. 6), we have,*

$$G_k = Correct(q,s) + \alpha \cdot \sum_{k'=k}^{K} r(q,p^{(k')}) - \beta \cdot \sum_{k'=k}^{K} \log \frac{\pi_\theta(s^{(k')}|q,p^{(k'-1)})}{\pi_{ref}(s^{(k')}|q,p^{(k'-1)})})$$

$$= Correct(q,s) + \alpha \cdot \sum_{k'=k}^{K-1}(r_{process}(q,p^{(k')}) - r_{process}(q,p^{(k'+1)} \cdot \mathbb{I}[k+1 < K])$$

$$- \beta \cdot \sum_{k'=k}^{K} \log \frac{\pi_\theta(s^{(k')}|q,p^{(k'-1)})}{\pi_{ref}(s^{(k')}|q,p^{(k'-1)})})$$

$$= Correct(q,s) + \alpha \cdot \left(\sum_{k'=k}^{K-1}(r_{process}(q,p^{(k')}) - \sum_{k'=k+1}^{K-1}(r_{process}(q,p^{(k')})\right)$$

$$- \beta \cdot \sum_{k'=k}^{K} \log \frac{\pi_\theta(s^{(k')}|q,p^{(k'-1)})}{\pi_{ref}(s^{(k')}|q,p^{(k'-1)})})$$

$$= Correct(q,s) + \alpha \cdot r_{process}(q,p^{(k)}) \cdot \mathbb{I}[k < K] - \beta \cdot \sum_{k'=k}^{K} \log \frac{\pi_\theta(s^{(k')}|q,p^{(k'-1)})}{\pi_{ref}(s^{(k')}|q,p^{(k'-1)})})$$

*Therefore,*

$$\nabla_\theta J_r(\pi_\theta) = \mathbb{E}_{q\sim\mathcal{D},s\sim\pi_\theta(\cdot|q)}[\sum_{k=1}^{K}\nabla_\theta \log \pi_\theta(s^{(k)}|q,p^{(k-1)}) \cdot (\alpha \cdot r_{process}(q,p^{(k)}) \cdot \mathbb{I}[k < K] \quad (12)$$

$$+ Correct(q,s) - \beta \cdot \sum_{k'=k}^{K} \log \frac{\pi_\theta(s^{(k')}|q,p^{(k'-1)})}{\pi_{ref}(s^{(k')}|q,p^{(k'-1)})})] \quad (13)$$

**Lemma 2 (Policy gradient of RL training with SR)** *Following (Sutton, 2018), the vanilla policy gradient of using the sparse success reward only is given by,*

$$\nabla_\theta J_r(\pi_\theta) = \mathbb{E}_{q\sim\mathcal{D},s\sim\pi_\theta(\cdot|q)}[\sum_{k=1}^{K}\nabla_\theta \log \pi_\theta(s^{(k)}|q,p^{(k-1)}) \cdot (Correct(q,s) \quad (14)$$

$$- \beta \cdot \sum_{k'=k}^{K} \log \frac{\pi_\theta(s^{(k')}|q,p^{(k'-1)})}{\pi_{ref}(s^{(k')}|q,p^{(k'-1)})})] \quad (15)$$

**Proof 2** *Omitted.*

## E.2 PRM AS VALUE

In this section, we are going to show the connection between PRM training and Q learning. We will show that PRMs can be interpreted as Q functions or value functions.

**Notation.** To train a PRM, the following elements are required,

- **A PRM** $r_{process}(q, p)$ where $q$ is the question and $p$ is a partial solution prefix.

- **A prompt dataset** of pairs of questions and partial solution prefices $\mathcal{D}_{prompt} = \{(q_i, p_i)\}_{i \in [N]}$.

- **A completer** $\pi_c$ that is a policy used for generating the full solution $s$ given a question $q$ and a partial solution prefix $p$. The generated full solution $s$ contains $p$ as the prefix and contain necessary rationales that lead to the final answer. The completer is not necessary an LLM policy and can also be a perfect oracle policy, or an LLM-based search policy.

- **A labeled dataset** constructed by using the completer $\pi_c$ to find solution for the prompt dataset $\mathcal{D}_{prompt}$, i.e. $\mathcal{D}_{label} = \{(q_i, p_i, s_i, y_i)|s_i \sim \pi_c(\cdot|q_i, p_i), y_i = \text{Correct}(q_i, s_i)\}$ where $s_i$ is a full solution generated by the completer $\pi_c$ and $y_i$ denotes the correctness label. The labels thus depend on the strength of the completer $\pi_c$.

- **A loss function** $\mathcal{L}$ for PRM training. By default, we use the logistic loss

$$\mathcal{L}(r_{process}, \mathcal{D}_{label}) = \mathbb{E}_{q_i, p_i, s_i, y_i \sim \mathcal{D}_{label}}[-y_i \log r_{process}(q_i, p_i) \tag{16}$$
$$- (1 - y_i) \log(1 - r_{process}(q_i, p_i))] \tag{17}$$

We first list some possible options for the completer,

- **A perfect oracle completer** $\pi_{orcale}$. We assume the existence of a perfect oracle completer $\pi_{oracle}$ that is perfectly rational in the sense that, given a question $q$ and a partial solution prefix $p$, $\pi_{orcale}$ would make optimal reasoning towards the correct answer while also not driving any conclusions conflicting to the partial solution prefix when completing the rest reasoning trajectory. Therefore, if the solution prefix $p$ contains any erroneous reasoning steps that conflict with the correct answer, $\pi_{oracle}$ can not identify the correct answer and thus the label $y$ for this pair $(q, p)$ would be zero.

- **A base LLM completer** $\pi_{base}$. $\pi_{base}$ is essentially an LLM that samples the rest of the solution given a question $q$ and a partial solution prefix $p$. We assume a temperature of 1.0 and do not consider any sampling techniques such as top-p and top-k sampling. Therefore the label $y$ is 1 with the probability of $\pi_{base}$ sampling a correct solution.

- **A LLM-based search completer** $\pi_{search}$. We specially consider the search approach for automatic process label generation in Wang et al. (2024b), which tries to find a correct solution out of the sampled $M$ solutions from a base LLM $\pi_{base}$. Clearly the search completer $\pi_{search}$ is an augmented policy of $\pi_{base}$. The probability of $\pi_{search}$ generating a correct solution is higher than the probability of $\pi_{base}$ generating a correct solution.

**Definition 1 (Value of a completer)** *Given a question $q$ and a solution prefix $p$, the value of a completer $\pi_c$ is defined as*

$$V_{\pi_c}(q, p) = \mathbb{E}_{s \sim \pi(\cdot|q, p)}[Correct(q, s)] \tag{18}$$

**Lemma 3 (PRM is learning a value function)** *An optimal PRM that achieves the lowest PRM training loss in Eq. 16 is equivalent to the value function of the completer.*

**Proof 3** *The optimal PRM $r^*_{process}(q_i, p_i)$ of PRM training loss in Eq. 16 is,*

$$r^*_{process}(q_i, p_i) = Pr[y_i = 1 | s_i \sim \pi_c(q_i, p_i), y_i = Correct(q_i, s_i)]$$

$$= Pr[Correct(q_i, s_i) | s_i \sim \pi_c(q_i, p_i)]$$

$$= E_{s_i \sim \pi_c(\cdot|q_i, p_i)}[Correct(q_i, s_i)]$$

$$= V_{\pi_c}(q_i, p_i)$$

*which is the same as the value of the completer $\pi_c$ in Eq. 18.*

Therefore PRM is actually learning the value of the completer $\pi$. In that sense, RL training that combines PR-Delta as dense rewards and the success reward is actually training the LLM to find correct solutions under the guidance of a stronger policy, i.e. the completer used to label the training data of the PRM.

**Corollary 1 (PR-Delta guides LLM reasoning with a stronger policy)** *Combining Lemma. 1 and Lemma. 3, PR-Delta actually guides the LLM to learn reasoning skills under the guidance of the stronger completer policy $\pi_c$, as shown in the policy gradient,*

$$\nabla_\theta J_r(\pi_\theta) = \mathbb{E}_{q \sim \mathcal{D}, s \sim \pi_\theta(\cdot|q)}[\sum_{k=1}^{K} \nabla_\theta \log \pi_\theta(s^{(k)}|q, p^{(k-1)}) \cdot (\alpha \cdot V_{\pi_c}(q, p^{(k)}) \cdot \mathbb{I}[k < K] \quad (19)$$

$$+ Correct(q, s) - \beta \cdot \sum_{k'=k}^{K} \log \frac{\pi_\theta(s^{(k')}|q, p^{(k'-1)})}{\pi_{ref}(s^{(k')}|q, p^{(k'-1)})})] \quad (20)$$

# F ADDITIONAL CASE STUDIES OF PRM

We evaluate the score of a open-sourced PRM 'ScalableMath/llemma-7b-prm-prm800k-level-1to3-hf'[3] on the following solution:

---

**Question**

What is the range of all the roots of $|x^2 - 6| = x$?

**Solution**

Step 1: Establish the equation by squaring both sides. We get $(x^2 - 6)^2 = x$.

PRM reward: 0.4874

Step 2: Expand the squared term. We get $(x^2 - 6)^2 = x^4 - 12x^2 + 36$.

PRM reward: 0.1667

Step 3: Set the expanded equation equal to $x$. We get $x^4 - 12x^2 + 36 = x$.

PRM reward: 0.4116

Step 4: Move all terms to one side to set the equation to zero. We get $x^4 - 12x^2 - x + 36 = 0$.

PRM reward: 0.7827

Step 5: Factor the polynomial. We can factor it as $(x^2 - 6)(x^2 + 1) = x$.

PRM reward: 0.0558

Step 6: Solve each factor for $x$. For $x^2 - 6 = x$:

$$x^2 - x - 6 = 0$$

$$(x - 3)(x + 2) = 0$$

---

[3]https://huggingface.co/ScalableMath/llemma-7b-prm-prm800k-level-1to3-hf

So, the solutions are $x = 3$ and $x = -2$. For $x^2 + 1 = x$:

$$x^2 - x + 1 = 0$$

This quadratic equation has no real roots because its discriminant is negative:

$$\Delta = (-1)^2 - 4 \cdot 1 \cdot 1 = 1 - 4 = -3$$

PRM reward: 0.4244

Step 7: Determine the range of the roots. The roots we found are $x = 3$ and $x = -2$.

PRM reward: 0.6234

Step 8: Identify the minimum and maximum values of the roots. The minimum value is $-2$ and the maximum value is 3. Therefore, the range of all the roots of $|x^2 - 6| = x$ is from $-2$ to 3. The answer is $\boxed{[-2, 3]}$.

PRM reward: 0.4244

