# OpenReview forum: "On Designing Effective RL Reward at Training Time for LLM Reasoning"
_ICLR.cc/2025/Conference — Submitted to ICLR 2025_

### Official Review · Reviewer_ffHa · 2024-11-01

**Soundness:** 2
**Presentation:** 2
**Contribution:** 2
**Rating:** 3
**Confidence:** 4

**Summary:**

The paper explores how to design effective RL rewards models to improve LLMs in mathematical tasks.
It evaluates two popular reward models - ORM and PRM - during RL training.
The authors argue that using these reward models do not improve and even degrade performance during RL training; due to "reward hacking".
The authors propose two ad-hoc reward refinement techniques, Clipping and Delta, and demonstrated their performance improvement.

**Strengths:**

- The paper has positioned itself well as an NLP paper by referring a number of recent papers on LLM and reward models.
- The solution suggested by the paper effectively improves from previous baseline.

**Weaknesses:**

- After indroducing ORM, the paper does nothing about it; since it is introduced and evaluated, we need at least an analysis on why it does not help.

- While the paper is suspecting the observed phenomenon as "reward hacking", I would say this is closer to the wrong usage of reward functions. As we have binary correctness label, and ORM and PRM try to get estimation of it with partial generation, all rewards will be nonnegative. In episodic settings without any discount, this naturally lead to the agents that favor long trajectory, i.e., long solution generation, regardless of whether its correct or not. The paper does not tell us about the hyperparameter choice of $\alpha$, but with some large enough $\alpha$, this may lead to agent preferring long wrong generations over short correct generations.

- The proposed method is, in my perspective, not novel. To solve sparse reward problems, the first go-to should be reward-shaping methods. the proposed delta mechanism can be understood as a potential-based reward shaping method where PRM is the potential function. The proposed clip-delta mechanism can be understood as a potential-based reward shaping method where PRM-clip is the potemtial function. The overall paper can actually be understood as using a reward shaping methods to handle sparse rewards of RLHF, and I believe the paper should position itself well against reward-shaping methods that have been extensively studied in the past.

- The paper argues that PR-Normed approach overfits and the performance degradation is severe. To see SR+proposed method algorithms do not suffer from the same problem, we need more information, e.g., learning curves of all algorithms.

- While in this paper the Delta (giving agent no preference on length of generation) or the Clip (giving agent preference on shorter generation) worked, it depends on the task; on the tasks where longer generations are advantageous, both modifications might not be beneficial. In that sense, the paper only evaluates on single task, and it would not be a robust assessment on the algorithm's performance.

**Questions:**

- Why does ORM not suffer from the same problem?
- What's the difference between PRM and ORM: why PRM does help when ORM does not help?
- If we use PR-Clip itself, then it becomes an algorithm that prefers shorter generation as we increase $\eta$. Do you have results on the performance on varying $\eta$?

---

> ### Author Response · Authors · 2024-11-21
> **Response by Authors (Part I)**
>
> We sincerely appreciate the reviewer for providing valuable feedback. We hope the following response can address the concerns raised by the reviewer.
>
> ## 1. On the Novelty of the Clip and Delta Mechanisms
> > “The proposed method is, in my perspective, not novel. To solve sparse reward problems, the first go-to should be reward-shaping methods. ”
> - We emphasize that, though reward shaping is a common approach in RL, it is challenging to design proper rewards to promote better reasoning skills, as shown by our in-depth analysis in Fig. 3 (also [available here](https://i.postimg.cc/Gp1Cq89Y/case-study.jpg)).
> - We conduct additonal case studies (Fig. 3) and theoretical analysis (Appendix E) on why the proposed methods can effectively utilize the PRM to enhance reasoning through RL training. Beyond their practical effectiveness, our findings provide valuable insights for the community on how to design effective RL rewards for reasoning tasks.
> - **Please refer to the Global Response and the revised paper for more details.**
>
> ## 2. Explanation of the Delta Mechanism and Connection to Potential-based Reward Shaping
> > “The proposed delta mechanism can be understood as a potential-based reward shaping method where PRM is the potential function. ... the paper should position itself well against reward-shaping methods that have been extensively studied in the past."
> - It is insightful to point out the connection between the delta mechanism and potential-based reward shaping methods. However, we clarify that **the Delta mechanism is NOT a potential-based reward shaping (PBRS) method** because the Delta mechanism does not fit in the mathematical form of PBRS.
> - By the definition of potential-based reward shaping in [1], a potential-based shaping function takes the form $F(s,a,s')=\gamma \Phi(s')-\Phi(s)$ for transition $s,a,s'$. However, the Delta mechanism uses $r(q,p^{(k)})=r_{process}(q,p^{(k)})-r_{process}(q,p^{(k+1)})$ and takes the form $F(s,a,s')=\Phi(s')-\Phi(s'')$ for transition $s,a,s',a',s''$ by defining $s=(q,p^{(k-1)}),s'=(q,p^{(k)}), s''=(q,p^{(k+1)}),\Phi(s')=r(q,p^{(k)}),\Phi(s'')=r(q,p^{(k+1)})$.
> - To better interpret the Delta mechanism, we provide additional theoretical analysis in Appendix E. We show the policy gradient of RL training with SR+PR-Delta can be given in a step-wise manner,
>     $$
>         \nabla_\theta J_{r}(\pi_\theta)=\mathbb E_{q\sim \mathcal D,s\sim \pi_\theta(\cdot|q)}\large[\nabla_\theta \log\pi_\theta(s|q)\cdot \text{Correct}(q,s)+ \alpha\cdot\underbrace{\sum_{k=1}^{K-1}\nabla_\theta \log\pi_\theta(s^{(k)}|q,p^{(k-1)})\cdot r_{process}(q, p^{(k)})}_{\text{Effect of the Delta mechanism}}\large] +\text{KL term}
>     $$
>     Thus, **the Delta mechanism enhances the single-step PRM rewards**.
> - We have added the discussion on related reward shaping methods in Sec 2.
>
>
> [1] Ng, Andrew Y., Daishi Harada, and Stuart Russell. "Policy invariance under reward transformations: Theory and application to reward shaping." Icml. Vol. 99. 1999.

---

> ### Author Response · Authors · 2024-11-21
> **Response by Authors (Part II)**
>
> ## 3. Analysis of PR \& Reward Shaping Coefficient for SR+PR
> > “While the paper is suspecting the observed phenomenon as "reward hacking", I would say this is closer to the wrong usage of reward functions.”
> - We agree with the reviewer that “reward hacking” is an inaccurate description. We have revised the description of the phenomenon in SR+PR training as “training collapse”.
> - We also agree that simply using PRM as dense rewards (PR) could be a wrong usage of the rewards. In our additional case studies, as shown in Fig. 3 (also [available here](https://i.postimg.cc/Gp1Cq89Y/case-study.jpg)), we identify the reward misspecification issue of PR. The proposed Delta mechanism can ensure the steps promoted by RL training are aligned with the PRM by optimizing single-step PRM rewards.
>
> > "The paper does not tell us about the hyperparameter choice of $\alpha$, but with some large enough $\alpha$, this may lead to agent preferring long wrong generations over short correct generations."
>
> - For SR+PR, we conduct an ablation study on the reward shaping coefficient $\alpha$. When $\alpha$ is small, though the test accuracy of SR+PR can be marginally higher than or on par with SR in early training epochs, SR+PR only achieves sub-optimal final accuracy. As we increase $\alpha$, training collapse happens, and the test accuracy gets worse. The results are given in Appendix A.3 and also shown in the following table.
>
> **Test accuracy of SR vs. SR+PR with different reward shaping coefficient $\alpha$ across training epochs:**
> |          | Epoch 1 | Epoch 2 | Epoch 3 | Epoch 4 | Epoch 5 |
> | -------- | ------- | ------- | ------- | ------- | ------- |
> | SR       | **29.26** |  29.72  |  29.86  | 30.16    |  **30.58** |
> | SR+PR ($\alpha=0.02$) | 29.25 | **30.00** | **29.88** |  **30.22** | 30.08|
> | SR+PR ($\alpha=0.05$)  | 21.9 | 18.92 | / | / | / |
> | SR+PR ($\alpha=0.1$)  | 14.10 | / | / | / | / |
> | SR+PR ($\alpha=0.2$)  | 11.16 | / | / | / | / |
>
> where we stop training after training collapse is observed. Tested on MATH test set.
>
> - Regardless of the value of $\alpha$, PR has some fundamental issues, as pointed out in our case studies in Fig. 3. Therefore, simply using PRM rewards as dense rewards could not reliably help enhance the reasoning skills of the LLM.
>
>
> ## 4. Comparison Between ORM and PRM
>
> > "After introducing ORM, the paper does nothing about it; since it is introduced and evaluated, we need at least an analysis on why it does not help."
> - We would like to clarify for the inappropriate description that ORM does not help RL training in the submission version. Actually, introducing ORM improves the sample efficiency but not significantly improve the final accuracy.
> - We hypothesize this is because the training targets of ORM and the critic of PPO with success rewards in the last token are equivalent. Therefore, introducing ORM offers a better initialization for the last-token value. However, the benefit of ORM would diminish when sufficient RL training is conducted.
>
> **Test Greedy accuracy of SR vs. SR+OR across training epochs:**
> |          | Epoch 1 | Epoch 2 | Epoch 3 | Epoch 4 | Epoch 5 |
> | -------- | ------- | ------- | ------- | ------- | ------- |
> | SR       | 29.26   | 29.72   | 29.86   | 30.16   | **30.58** |
> | SR+OR | **29.32** | **30.04** | **30.08** | **30.48** | 30.57  |
>
> **Test accuracy of SR vs. SR+OR:**
> |          | Greedy | Sampling |
> | -------- | ------- | ------- |
> | SR       | **30.58** | 27.05   |
> | SR+OR | 30.57 | **27.12** |
>
> Both are tested on MATH test set.
>
> - We have updated the analysis of ORM in the revised version.
>
> > "What's the difference between PRM and ORM"
>
> There are two main differences between PRM and ORM:
> - Training approach: PRM is trained to predict the correctness of a step. ORM is trained to predict the probability of the final correctness.
> - Density of reward signals: OR uses ORM to provide sparse rewards that only occur at the end of the solution. PRM provides step-level dense rewards in PR.
>
> > "Why does ORM not suffer from the same problem?"
> - The sparse rewards provided by ORM in the range of [0,1] naturally ensure a bounded RL objective. Therefore, training stability can be easily achieved.
>
> > "Why PRM does help when ORM does not help?"
> - PRM provides dense training signals at the reasoning step level rather than just at the solution level. This enables better credit assignment during training, helping the LLM to correct reasoning errors more effectively.
> - The effect of PRM can be better illustrated in the right case of Fig. 3. In this case, when the final answer of a solution is incorrect and the success reward is zero, PRM provides guidance on which steps are sub-optimal and should be avoided.

---

> ### Author Response · Authors · 2024-11-21
> **Response by Authors (Part III)**
>
> ## 5. Training Curves of PR-Normed and the Proposed Methods
> > "The paper argues that PR-Normed approach overfits and the performance degradation is severe. To see SR+proposed method algorithms do not suffer from the same problem, we need more information, e.g., learning curves of all algorithms."
> - During early training epochs, PR-Normed shows a sign of overfitting and only achieves sub-optimal test accuracy compared with SR, as shown in the following table. RL training of SR+PR-Normed also suffers from significant performance degradation after Epoch 3.
>
> **Train/Test accuracy of SR vs. SR+PR-Normed across training epochs:**
> | train acc./ test acc. | Epoch 1 | Epoch 2 | Epoch 3 | Epoch 4 | Epoch 5 |
> | -------- | ------- | ------- | ------- | ------- | ------- |
> | SR       | 30.54 / **29.26** | 34.29 / **29.72** | **35.16** / **29.86** | **38.52** / **30.16** | **38.07** / **30.58** |
> | SR+PR-Normed | **32.23** / 28.68 | **36.13** / 29.66 | 34.79 / 25.9  | 23.18 / 9.8  | 25.39 / 12.36  |
>
>
> where "a/b" denotes training accuracy $a\%$ and test greedy accuracy $b\%$. Tested on MATH test set.
>
>
> - To address your concern, we have provided the training curves of all algorithms in Appendix A.2 (Fig. 7 ~ Fig.11, also [available here](https://i.postimg.cc/jd1HPLQM/training-curves.jpg)).
>
>
> ## 6. Additional Ablation Study of the Reward Threshold $\eta$
>
> > "If we use PR-Clip itself, then it becomes an algorithm that prefers shorter generations as we increase $\eta$. Do you have results on the performance on varying $\eta$?"
>
> - We thank the reviewer for requesting the ablation study of $\eta$.
> - In our experiments, by default we set $\eta$ to be the average value of PRM rewards of all reasoning steps related to one question in a training batch. This choice could avoid explicitly tuning the optimal $\eta$ for different PRMs.
> - We conduct ablation study of the reward threshold $\eta$ used in PR-Clip, as shown in Fig. 12 of Appendix A.2 and in the following table. Surprisingly, we find a fixed $\eta$ brings more stable improvements and $\eta=0.7$ obtains the best performance and can surpass RL training with SR.
>
> **Ablation study of $\eta$:**
> |          | Epoch 1 | Epoch 2 | Epoch 3 | Epoch 4 | Epoch 5 |
> | -------- | ------- | ------- | ------- | ------- | ------- |
> | SR       | 29.26 |  29.72  |  29.86  | 30.16    |  30.58 |
> | SR+PR-Clip ($\eta=0.2$) | 29.68 | 30.4 | 30.5 | 30.28 | 30.78 |
> | SR+PR-Clip  ($\eta=0.7$)  | 29.4 | 30.46 | **30.44** | **30.84** | **30.98** |
> | SR+PR-Clip  ($\eta=$ mean)  | **29.84** | **30.56** | 29.62 | 30.62 | 30.3 |
>
> Tested on MATH test set. '$\eta=$ mean' means that we set $\eta$ to be the average value of PRM rewards of all reasoning steps related to one question in a training batch.
>
> ## 7. Discussion on Preference of Generation Length
> > “While in this paper the Delta (giving agent no preference on length of generation)  or the Clip (giving agent preference on shorter generation) worked, it depends on the task; on the tasks where longer generations are advantageous, both modifications might not be beneficial. In that sense, the paper only evaluates on single task, and it would not be a robust assessment on the algorithm's performance.”
> - We emphasize that the Clip and the Delta mechanisms are proposed to tackle the issues of simply applying PRM as reward shaping as analyzed in Sec. 4. The significant increment of length is a consequence of one of the identified issues. Please refer to Sec. 3 for more details.
> - We suggest combining both the Clip and the Delta mechanisms in practice, which mitigates issues from different aspects and would not explicitly introduce length biases. Regardless of the generation length, as long as PRM can evaluate the quality of reasoning steps accurately, RL training could improve the accuracy of the LLM.
> - We also argue that there is no general preference for the generation length in mathematical reasoning tasks. The optimal generation length often depends on the problem at hand. While some problems may benefit from detailed, step-by-step solutions to ensure clarity and accuracy, others might be better solved with concise, high-level reasoning. As such, there is no universal preference for shorter or longer generations, as both can be equally valid depending on the problem's requirements.
>
> We hope our response addresses your concerns, and we welcome any further questions or suggestions.

---

### Official Review · Reviewer_sMoF · 2024-11-01

**Soundness:** 2
**Presentation:** 3
**Contribution:** 1
**Rating:** 3
**Confidence:** 4

**Summary:**

This paper introduces two methods to prevent reward hacking while training LLM with reinforcement learning algorithms, Clipping and Delta. The paper shows superior results when adding these techniques on top of process rewards, compared to baselines without these techniques.

**Strengths:**

The paper is written clearly, and well-motivated. Extensive experiments are done, ablating different parts of the algorithm to show that clipping and delta mechanisms are actually helping.

**Weaknesses:**

Reward clipping is not a novel idea and has been studied before. Reward hacking, even more so. IMO, a valid baseline for this paper needs to be a well-written PRM. However, looking at the baselines used in this paper, whenever PR is used the result is worse than SR alone. Other papers have already shown PRM is better than ORM alone. Clearly, the baseline is too weak and isn't setup correctly.

Basically, the authors are showing evidence that SR + PR + reward-hacking-prevention is better than SR alone, or SR + plain PR. I don't find this to be a new contribution.

The work also lacks direct examination of how these two methods work under the hood. For example, how often is the reward non-zero after clipping? i.e. Is this truly a dense reward? When delta is applied on top of clipping, the argument that the reward is bound no longer holds, because r_process(q, p_k+1) can be 0. So what's really the mechanism in this case?

In Table 2, the experimental results show pretty marginal (or, non-existent) improvements on greedy decoding, on the only valid RL baseline, Qwen2.5-Math-7B-Instruct. The other comparisons are pretty meaningless since we all know RL works & it's not the contribution of this paper.

**Questions:**

How often is the reward non-zero after clipping? i.e. Is this truly a dense reward?

Why does none of the PRM method in baselines work?

---

> ### Author Response · Authors · 2024-11-21
> **Response by Authors (Part I)**
>
> We sincerely thank the reviewer for the detailed feedback to help us improve the quality of our work. We hope the following response can address the points raised by the reviewer.
>
> ## 1. Position of Our Work
> > “Other papers have already shown PRM is better than ORM alone.”
>
> We emphasize that, to the best of our knowledge, though PRM is shown to be effective for test-time search for LLM reasoning, **it is still unclear whether ORM and PRM can bring additional benefits to RL training with success rewards.** Prior to our work, [1], [2], [3] and [4] apply ORM/PRM as reward shaping for RL training, but the actual effects of ORM and PRM are unclear due to a lack of sufficient ablation study. **Our work also presents the first thorough study on the issues of simply adopting PRM as reward shaping in RL training.**
>
> [1] Shao, Zhihong, et al. "Deepseekmath: Pushing the limits of mathematical reasoning in open language models." arXiv preprint arXiv:2402.03300 (2024).
>
> [2] Yang, An, et al. "Qwen2. 5-math technical report: Toward mathematical expert model via self-improvement." arXiv preprint arXiv:2409.12122 (2024).
>
> [3] Wang, Peiyi, et al. "Math-shepherd: Verify and reinforce llms step-by-step without human annotations." arXiv preprint arXiv:2312.08935 (2023).
>
> [4] Havrilla, Alex, et al. "Teaching large language models to reason with reinforcement learning." ICML 2024 Workshop AI4MATH.
>
> ## 2. On the Novelty of Our Work
> > “Basically, the authors are showing evidence that SR + PR + reward-hacking-prevention is better than SR alone, or SR + plain PR. I don't find this to be a new contribution.” “Reward clipping is not a novel idea and has been studied before. Reward hacking, even more so.”
>
> - We emphasize that, besides preventing training collapse, the mechanisms can effectively utilize the PRM to promote better reasoning skills in RL training.
> - We conduct additional case studies in Sec. 4 and theoretical analysis in Appendix E to better illustrate the motivation and the effect of the proposed methods.
> - Although the proposed methods are simple, our in-depth analysis in Sec. 4 reveals why they are effective for PRM in RL training. Our analysis presents valuable insights to the community on how to design effective RL rewards for LLM reasoning.
> -  **Please refer to the Global Response and the revised paper for more details.**
>
> ## 3. Performance of PRM-based Baselines
> > “However, looking at the baselines used in this paper, whenever PR is used the result is worse than SR alone. …. Clearly, the baseline is too weak and isn't setup correctly.” “Why does none of the PRM method in baselines work?”
>
> We would like to clarify that baselines are set up correctly. PRM-based baselines can improve RL training in training accuracy or test accuracy but can only achieve sub-optimal final performance. Training curves of all methods are provided in Appendix A.2.
>
> Here we discuss the specific performance of baselines.
> - **PR-Normed**: During early training epochs, PR-Normed shows a sign of overfitting and only achieves sub-optimal test accuracy compared with SR, as shown in the following table. RL training of SR+PR-Normed also suffers from significant performance degradation after Epoch 3.
>
> **Train/Test accuracy of SR vs. SR+PR-Normed across training epochs:**
> | train acc./ test acc. | Epoch 1 | Epoch 2 | Epoch 3 | Epoch 4 | Epoch 5 |
> | -------- | ------- | ------- | ------- | ------- | ------- |
> | SR       | 30.54 / **29.26** | 34.29 / **29.72** | **35.16** / **29.86** | **38.52** / **30.16** | **38.07** / **30.58** |
> | SR+PR-Normed | **32.23** / 28.68 | **36.13** / 29.66 | 34.79 / 25.9  | 23.18 / 9.8  | 25.39 / 12.36  |
>
> where "a/b" denotes training accuracy $a\%$ and test greedy accuracy $b\%$. Tested on MATH test set.
>
>
> - For **SR+PR**, we conduct an ablation study on the reward shaping coefficient $\alpha$. When $\alpha$ is small, SR+PR only achieves sub-optimal final accuracy. As we increase $\alpha$, training collapse happens, and the test accuracy gets worse. The results are shown in the following table.
> - We also note that, regardless of the value of $\alpha$, PR has some fundamental issues that may lead the LLM to learn undesired behavior patterns, as pointed out in our additional case studies in Fig. 3 of Sec. 4 (also [available here](https://i.postimg.cc/Gp1Cq89Y/case-study.jpg)).
>
> **Test accuracy of SR vs. SR+PR with different reward shaping coefficient $\alpha$ across training epochs:**
> |          | Epoch 1 | Epoch 2 | Epoch 3 | Epoch 4 | Epoch 5 |
> | -------- | ------- | ------- | ------- | ------- | ------- |
> | SR       | **29.26** |  29.72  |  29.86  | 30.16    |  **30.58** |
> | SR+PR ($\alpha=0.02$) | 29.25 | **30.00** | **29.88** |  **30.22** | 30.08|
> | SR+PR ($\alpha=0.05$)  | 21.9 | 18.92 | / | / | / |
> | SR+PR ($\alpha=0.1$)  | 14.10 | / | / | / | / |
> | SR+PR ($\alpha=0.2$)  | 11.16 | / | / | / | / |
>
> where we stop training after training collapse is observed. Tested on MATH test set.

---

> ### Author Response · Authors · 2024-11-21
> **Response by Authors (Part II)**
>
> ## 4. The Effect of the Clip and Delta Mechanisms
> > “The work also lacks direct examination of how these two methods work under the hood. For example, how often is the reward nonzero after clipping? i.e. Is this truly a dense reward? When delta is applied on top of clipping, the argument that the reward is bound no longer holds, because r_process(q, p_k+1) can be 0. So what's really the mechanism in this case?”
>
> - **Clip mechanism**: In our experiments, we set $\eta$ to be the average value of PRM rewards of all reasoning steps related to one question in a training batch. After checking the training statistics, we find the ratio of nonzero clipped PRM rewards to be around $50\%$, showing that the Clip mechanism **indeed provides dense rewards for RL training.** The clip ratio during training is provided in Appendix A.2.
> - **PR-Clip-Delta**: Regarding PR-Clip-Delta, by Eq. 5, the Clip mechanism first produces a reward $r_{PR-Clip}(q,p^{(k)})=min(r_{process}(q,p^{(k)})-\eta,0)\in[-1,0]$ since $r_{process}(q,p^{(k)})\in[0,1]$. For PR-Clip–Delta, by Eq. 6, the Delta mechanism generates $r_{PR-Clip-Delta}(q,p^{(k)})=r_{PR-Clip}(q,p^{(k)})-r_{PR-Clip}(q,p^{(k+1)})\in[-1,1]$ for $k<K-1$, $r_{PR-Clip-Delta}(q,p^{(k)})=r_{PR-Clip}(q,p^{(k)})\in[-1,0]$ for $k=K-1$, and $r_{PR-Clip-Delta}(q,p^{(K)})=0$. The return of PR-Clip-Delta is $\alpha\cdot r_{PR-Clip}(q,p^{(k)})+\text{Correct}(q,s)$ for intermediate steps and **therefore both the return and the reward of PR-Clip-Delta are bounded**.
>
> ## 5. Comparison between SR and SR+PR-Clip-Delta Across All Evaluated LLMs & The Improvement on Qwen2.5-Math-7B-Instruct
> > “In Table 2, the experimental results show pretty marginal (or, non-existent) improvements in greedy decoding, on the only valid RL baseline, Qwen2.5-Math-7B-Instruct. The other comparisons are pretty meaningless since we all know RL works & it's not the contribution of this paper.”
>
> - We clarify that our experiments present a comparison between SR and SR+PR-Clip-Delta across all evaluated LLMs in Fig. 5 of Sec. 5(also [available here](https://i.postimg.cc/qRTLfCDT/perf-improve.jpg)), where **using PR-Clip-Delta as dense rewards improves RL training with success rewards only across all evaluated LLMs**. We also apologize for the misleading effect caused by the absence of training results of PPO w. SR in the main table (Table 2). We have updated Table. 2 (also [available here](https://i.postimg.cc/v8xjwWZJ/main-table.jpg)) to explicitly include RL training results using success rewards for greater clarity.
> - **Improvement on Qwen2.5-Math-7B-Instruct:** On Qwen2.5-Math-7B-Instruct, adding PR-Clip-Delta as dense rewards also enhances the **sampling accuracy** and the **Pass@16** accuracy over RL training with success reward only. This indicates that the model learns better reasoning skills. Below is the detailed performance comparison:
>
>
> **Accuracy on MATH test set of SR vs. SR+PR-Clip-Delta on Qwen2.5-Math-7B-Instruct:**
> |          | Greedy | Sampling | Pass@16 |
> | -------- | ------- | ------- | ------- |
> | Qwen2.5-Math-7B-Instruct       | 83.3  | 52.76  | 86.6 |
> | +PPO w. SR | 83.16 | 79.95 | 92.46 |
> | +PPO w. SR+PR-Clip-Delta | **83.38** | **81.22** | **92.60** |
>
> - Our main focus is to understand how to unlock the potential of PRM in RL training without additional data. Further improving the most competitive LLMs would require techniques beyond RL and reward design, such as developing stronger PRMs and using additional data. These directions are beyond the scope of our work and we will leave them as future work. We believe our work could still provide guidance on designing rewards when stronger PRMs or more advanced RL algorithms are available.
>
> We hope our response addresses your concerns, and we welcome any further questions or suggestions.

---

### Official Review · Reviewer_6ViF · 2024-11-03

**Soundness:** 3
**Presentation:** 3
**Contribution:** 2
**Rating:** 5
**Confidence:** 3

**Summary:**

This paper investigates how to effectively use reward models during reinforcement learning (RL) training to improve LLMs' mathematical reasoning abilities. The authors discover that traditional reward models can either be ineffective (in the case of Outcome-supervised Reward Models) or lead to reward hacking through unnecessary repetition of steps (in the case of Process-supervised Reward Models). To address these issues, they introduce two novel techniques - "Clipping" and "Delta" mechanisms - that help prevent reward exploitation while maintaining the benefits of process rewards. Using these refined reward techniques, they demonstrate consistent improvements across various LLMs, including enhancing the performance of state-of-the-art models like Qwen2.5-Math-7B-Instruct on the MATH and GSM8K benchmarks.

The key innovation is showing that while reward models are useful for inference-time improvements, they need careful refinement to be effective during RL training. Their solutions help stabilize training and prevent the model from gaming the reward system through repetitive or unnecessary steps.

**Strengths:**

Strengths:
1. Identifying the issues with using rewards: The authors systematically analyze both Outcome-supervised Reward Models (ORM) and Process-supervised Reward Models (PRM), revealing important limitations of each approach. They demonstrate that ORMs, despite working well at inference time, don't provide additional benefits beyond success rewards during training. More significantly, they uncover a serious reward hacking issue with PRMs, where models learn to game the system by repeating simple or unnecessary steps to achieve high rewards. This analysis is particularly valuable because previous work primarily focused on using rewards at inference time, making this identification a novel contribution to the field.
2. Experiments: The authors conduct comprehensive evaluations across multiple model sizes (1.5B and 7B parameters) and variants (including Qwen2, Qwen2.5, and both general and math-specific models) on standard benchmarks like MATH and GSM8K. Their experimental design includes thorough ablation studies showing the impact of different components, careful comparisons of various reward mechanisms, and detailed analysis using synthetic examples to demonstrate reward hacking.

**Weaknesses:**

Limited Novelty:
- Clipping is indeed a well-established technique in the RL community, commonly used to stabilize training
- The concept of bounded rewards is a fundamental principle in RL, not a new innovation
- The Delta mechanism, while presented as novel, essentially implements reward shaping - another well-known concept in RL

The paper's contribution seems more incremental than innovative: It primarily applies known RL techniques to the specific context of LLM reasoning. The main finding that reward models can be exploited during training is somewhat expected given the general challenges of reward design in RL. The solutions proposed (Clipping and Delta mechanisms) are straightforward applications of existing RL principles.

Missing Reward Shaping related work:
1. Policy Invariance Under Reward Transformations: Theory and Application to Reward Shaping, Ng et.al
2. Hindsight credit assignment,  Harutyunyan et.al
3. Rudder: return decomposition for delayed rewards, Arjona et.al
4. Align-RUDDER: Learning from few demonstrations, Patil et.al
5. Modern Hopfield networks for return decomposition for delayed rewards, Widirich et.al

**Questions:**

My current criticism is regarding the novelty of the paper. I am willing to increase my score if the authors convince me that the clipping and delta are novel enough.

---

> ### Author Response · Authors · 2024-11-21
> **Response by Authors**
>
> We thank the reviewer for the valuable feedback and the interest in our work. Here we address the concerns raised by the reviewer:
>
> ## 1. Limited Novelty
> > “The paper's contribution seems more incremental than innovative: It primarily applies known RL techniques to the specific context of LLM reasoning. The main finding that reward models can be exploited during training is somewhat expected given the general challenges of reward design in RL. The solutions proposed (Clipping and Delta mechanisms) are straightforward applications of existing RL principles.”
>
> - We emphasize that the Clip and the Delta mechanisms are introduced to effectively the PRM to promote better reasoning skills, instead of simple applications of existing techniques.
> - We conduct additional case studies in Fig. 3 (also [available here](https://i.postimg.cc/Gp1Cq89Y/case-study.jpg)) and theoretical analysis in Appendix E to better illustrate the motivation and the effect of the proposed methods.
> - Our in-depth analysis in Sec. 4 reveals why they are effective for PRM in RL training. Beyond their practical effectiveness, our findings provide valuable insights for the community on how to design effective RL rewards for reasoning tasks.
> - **Please refer to the Global Response and the revised paper for more details.**
>
>
> ## 2. Missing Reward Shaping related work
>
> We thank the reviewer for pointing out the missed related work on reward shaping. We have added a discussion about the related works in the revised version.
>
>
> We hope our response addresses your concerns, and we welcome any further questions or suggestions.

---

### Official Review · Reviewer_gCaQ · 2024-11-03

**Soundness:** 3
**Presentation:** 3
**Contribution:** 3
**Rating:** 8
**Confidence:** 2

**Summary:**

This work discusses the role of reward models in enhancing the reasoning capability of LLM models during RL training, which is under-explored compared to inference. It shows the impact of popular reward models, the Outcome-supervised Reward Model (ORM) and the Processsupervised Reward Model (PRM), on the performance of LLMs after being combined with the success sparse reward signals on math problems. It was observed that such reward models may not help or even hurt the performance of the LLM due to the reward hacking issue. This work proposes two reward refinement methods to tackle this issue, named Clipping and Delta. Those techniques have shown potential in stabilizing the RL training of a collection of LLMs when evaluated on the MATH and GSM8K benchmarks. In addition, performance improvement across all evaluated LLMs can be obtained by carefully designing a reward function for pure RL training.

**Strengths:**

- The paper is well-written, and the problem is nicely motivated.
- The empirical results are enough to assess the potential of the reward model, with some techniques for mitigating reward hacking during RL training to enhance the LLM reasoning.
- I appreciate the case study shown in Fig. 2 and the others added in the appendix.

**Weaknesses:**

## Major Comments:
- We saw in the empirical results that the improvement in the small models was higher than in the larger ones. I disagree with the authors about the importance of evaluating the study or the proposed techniques on larger models.

## Minor Comments:
- Typo in Line 057: "on the reward models, it remains **un**clear whether the reward models can provide additional training".

**Questions:**

- As far as I understand, the outcome reward (OR) gives a likelihood that the solution is correct. That means a solution can have a success reward of zero but still be close to the correct solution; hence, the likelihood could be relatively high. It can also be high because of the uncertainty of the reward model. **Why do you still think that did not help when combined with the success reward?** I believe a probability of success in the form of reward, instead of ones and zeros, should contribute to the learning. If I understood the outcome reward wrong, then the explanation of the reward models was unclear.

---

> ### Author Response · Authors · 2024-11-21
> **Response by Authors**
>
> We thank the reviewer for the valuable feedback. Here we respond to the points raised by the reviewer,
>
> ## 1. Discussion on the Evaluation of Larger Models
> > "We saw in the empirical results that the improvement in the small models was higher than in the larger ones. I disagree with the authors about the importance of evaluating the study or the proposed techniques on larger models."
>
> - We believe that the evaluation of larger and stronger models can help us understand the limitations of our approach and identify directions to further improve the most competitive LLMs with RL training. Potential improvement directions include augmenting the training distribution and enhancing the quality of PRM. We also believe that our approach is applicable to stronger models when a better PRM is available. However, these directions are out of the scope of our work and we leave them as future work.
>
> ## 2. Effect of ORM in RL Training
> > "As far as I understand, the outcome reward (OR) gives a likelihood that the solution is correct. That means a solution can have a success reward of zero but still be close to the correct solution; hence, the likelihood could be relatively high. It can also be high because of the uncertainty of the reward model. Why do you still think that did not help when combined with the success reward? I believe a probability of success in the form of reward, instead of ones and zeros, should contribute to the learning. If I understood the outcome reward wrong, then the explanation of the reward models was unclear."
> - We would like to clarify for the inappropriate description that ORM does not help RL training in the submission version. Actually, introducing ORM improves the sample efficiency but not significantly improve the final accuracy. The data is given in the following table.
> - We agree with the reviewer that ORM can bring some benefits. Indeed the training targets of ORM and the critic of PPO with success rewards in the last token are equivalent. Therefore, introducing ORM offers a better initialization for the last-token value. However, the benefit of ORM would diminish when sufficient RL training is conducted.
>
>
> **Test Greedy accuracy of SR vs. SR+OR across training epochs:**
> |          | Epoch 1 | Epoch 2 | Epoch 3 | Epoch 4 | Epoch 5 |
> | -------- | ------- | ------- | ------- | ------- | ------- |
> | SR       | 29.26   | 29.72   | 29.86   | 30.16   | **30.58** |
> | SR+OR | **29.32** | **30.04** | **30.08** | **30.48** | 30.57  |
>
> **Test accuracy of SR vs. SR+OR:**
> |          | Greedy | Sampling |
> | -------- | ------- | ------- |
> | SR       | **30.58** | 27.05   |
> | SR+OR | 30.57 | **27.12** |
>
> Both are tested on MATH test set.

---

> > ### Comment · Reviewer_gCaQ · 2024-12-01
> > **Acknowledgment**
> >
> > Dear Authors,
> >
> > Thank you for your response! I will retain my score while also keeping my confidence score.

---

### Official Review · Reviewer_MaFz · 2024-11-04

**Soundness:** 3
**Presentation:** 3
**Contribution:** 3
**Rating:** 6
**Confidence:** 3

**Summary:**

The paper examines the use of learned reward models, such as Outcome-supervised (ORM) and Process-supervised Reward Model (PRM), to enhance reasoning in LLMs during RL training. They show that ORM does not improve and PRM can even hinder RL training due to reward hacking. The authors propose “Clip” and “Delta” to address this.

**Strengths:**

- The proposed idea is new, interesting, and well-motivated.
- The paper is easy to read and follow.
- The addressed problem is of significance.

**Weaknesses:**

- More details on the experimental setup could be provided for reproducibility, including the reward thresholds for the Clip and Delta mechanisms and hyperparameters for PPO.
- Smaller LLMs may offer a larger scope of improvement, and so the proposed methods may seem to have been successful. However, to confirm the advantage, experiments on larger LLMs may be necessary; e.g., the paper reports GPT-4o-2024-08-06’s performance to be 92.9 on GSM8K, which is higher than almost all other models and variations, and offers less scope of improvement.

**Questions:**

- Any discussion on the computational overhead or ease of integration of Clip and Delta into existing workflows would be beneficial.
- Although mathematical reasoning is a good testbed, any comments on the potential applicability of these techniques in non-mathematical or multi-modal reasoning tasks?

---

> ### Author Response · Authors · 2024-11-21
> **Response by Authors**
>
> We thank the reviewer for the valuable feedback and the interest in our work. Herein, we address the points raised by the reviewer:
>
>
> ## 1. Experiment Setup for Reproducibility
> > "More details on the experimental setup could be provided for reproducibility, including the reward thresholds for the Clip and Delta mechanisms and hyperparameters for PPO."
> - We have added the hyperparameters for PPO in Appendix D.
> - In our experiments, we set $\eta$ to be the average value of PRM rewards of all reasoning steps related to one question in a training batch. We implement the training pipeline based on ReaLHF [1] by implementing the success reward and the dense rewards provided by PRM.
>
> [1] Mei, Zhiyu, et al. "ReaLHF: Optimized RLHF Training for Large Language Models through Parameter Reallocation." arXiv preprint arXiv:2406.14088 (2024).
>
> ## 2. Experiments on Larger LLMs
> > "Smaller LLMs may offer a larger scope of improvement, and so the proposed methods may seem to have been successful. However, to confirm the advantage, experiments on larger LLMs may be necessary; e.g., the paper reports GPT-4o-2024-08-06's performance to be 92.9 on GSM8K, which is higher than almost all other models and variations, and offers less scope of improvement."
> - **Our main focus is investigate how to unleash the potential of PRM in RL training for LLM reasoning.** Our main experiments over a diverse set of LLMs in Sec. 4 present a comparison between RL training with success rewards only and RL training that combines PR-Clip-Delta and success rewards. We believe our evaluation presents a solid empirical justification for the effectiveness of the proposed methods.
> - If a more powerful PRM is available or additional data is available, we believe the proposed approaches in our work could also benefit stronger and larger models. However, these directions are out of the scope of our work and we leave them as future work.
>
> ## 3. Computational Overhead
> > "Any discussion on the computational overhead or ease of integration of Clip and Delta into existing workflows would be beneficial."
> - Both the Clip and the Delta mechanisms are straightforward to integrate into existing workflows.
> - The implementation of the Clip mechanism involves computing the mean of the reward as a threshold after reward calculation, followed by applying the formula specified in Eq. 5. This additional step is computationally lightweight and seamlessly fits within the existing reward processing pipeline.
> - The Delta mechanism requires computing the difference between rewards from two adjacent steps, a process that is both conceptually simple and computationally efficient. As such, neither method introduces significant overhead, ensuring their ease of adoption.
> - We have also added related discussion in Appendix D.
>
> ## 4. Extension to Other Tasks
> > "Although mathematical reasoning is a good testbed, any comments on the potential applicability of these techniques in non-mathematical or multi-modal reasoning tasks?"
>
> We thank the reviewer for raising the important question of broader applicability. While our work focuses on mathematical reasoning, the proposed techniques are not inherently limited to this domain. The approach can indeed be extended to other reasoning tasks, such as coding challenges or multi-modal reasoning, as long as two key conditions are met: (1) a task-specific partitioning of reasoning steps is feasible, and (2) a reliable success signal is available. We consider these applications promising directions for future exploration.
>
> We hope our response addresses your concerns, and we welcome any further questions or suggestions.

---

### Official Review · Reviewer_7bHH · 2024-11-04

**Soundness:** 4
**Presentation:** 3
**Contribution:** 3
**Rating:** 6
**Confidence:** 3

**Summary:**

This work has two important messages for the field of LLM Reasoning: 1) the rewards models, especially the PRM can be hackable. Therefore, integrating them into LLM RL training may lead to hacking this reward performing worse than just removing it. 2) The paper argues that by clipping and then delta-ing the PRM rewards one can actually use these rewards models to gain a boost in their performance.

**Strengths:**

I think the paper has important messages for LLM reasoning community:

1) The message that PRMs are hackable is important and valuable. Also, the paper digs into showing what goes wrong which provides insight into what actually goes wrong when using them. That the LLM trained with these PRMs leans towards some steps that are correct but does not move us closer to a solution. I think this contribution is also important.

 2) Also, the paper shows limiting these rewards is not obvious. It is only by mixing the clipping and the delta method that they can boost the performance. This is interesting that it shows the problem is serious, although I don't agree that the clip-delta completely solved it.

**Weaknesses:**

I think the paper focuses a lot on the boost it gets from mixing the clip and delta. However, I have some concerns if the clip and delta is a generalizable approach. First, the delta mechanism seems unmotivated. There is not nothing wrong with being unmotivated if it works super well. But, I think the gains are modest. The delta mechanism rewards action `a_t` if the reward of action `a_{t+1}` is less which is a very strong change to the RL environment. I understand that some interesting properties like the summation and small change to the returns arise from this, but I believe this is still an unmotivated change to the environment. However, maybe I don’t have the correct intuition on this. It is quite surprising that none of them work well on their own which makes me wonder if there is any intuition behind why this mixture of both works better? I am just afraid that this is not that well motivated.

**Questions:**

1- What is the motivation behind the delta mechansim? Why should we assign credit to a_t if the reward of a_{t+1} is lower?

2- Can we consider the gains by mixing the clip and delta mechanism are strong and not modest?

3- Your PRM is trained on automatic data. Is this possible that this hackability issue is because of the noise caused by the automatic procedure and would be avoided when trained on human-data or better PRM data generation (I understand better PRM data generation is itself a research question)?

4- Is there any intuition on why both clip-and-delta should be applied to get a boost and none of them work well in isolation?

---

> ### Author Response · Authors · 2024-11-21
> **Response by Authors**
>
> We thank the reviewer for the thorough assessment of our paper. Herein, we address the points raised by the reviewer:
> ## 1. Motivation and Explanation of the Delta Mechanism
> > "What is the motivation behind the delta mechanism? Why should we assign credit to a_t if the reward of a_{t+1} is lower?"
>
> - We conduct additional case studies and theoretical analysis to better illustrate the motivation and the effect of the proposed methods. **Please refer to the Global Response and the revised paper for more details.**
> - Our additional case studies in Fig. 3 (also [available here](https://i.postimg.cc/Gp1Cq89Y/case-study.jpg)) reveal the reward misspecification issue when directly using PRM rewards as dense rewards, which could make RL training mistakenly promote incorrect steps.
> - **The Delta mechanism ensures the steps promoted by RL training are aligned with the PRM.** The effect of the Delta mechanism can be better illustrated through our additional theoretical analysis of SR+PR-Delta in Appendix E. We show the policy gradient of RL training with SR+PR-Delta can be given in a step-wise manner,
>     $$
>         \nabla_\theta J_{r}(\pi_\theta)=\mathbb E_{q\sim \mathcal D,s\sim \pi_\theta(\cdot|q)}\large[\nabla_\theta \log\pi_\theta(s|q)\cdot \text{Correct}(q,s)+ \alpha\cdot\underbrace{\sum_{k=1}^{K-1}\nabla_\theta \log\pi_\theta(s^{(k)}|q,p^{(k-1)})\cdot r_{process}(q, p^{(k)})}_{\text{Effect of the Delta mechanism}}\large] +\text{KL term}
>     $$
>     Thus, **the Delta mechanism optimizes the single-step PRM rewards**.
>
> ## 2. Explanation of Why the Clip and Delta Mechanisms Can be Combined
> > "Is there any intuition on why both clip-and-delta should be applied to get a boost and none of them work well in isolation?"
>
> - We clarify that both the Clip and the Delta mechanisms can enhance RL training with success rewards. From the ablation study, as shown in Table. 1, both SR+PR-Clip and SR+PR-Delta attain better sampling accuracy compared with Success Reward. By further applying the Delta mechanism over PR-Clip, PR-Clip-Delta brings a stable and precise improvement over SR.
> - The proposed techniques are introduced to mitigate the identified issues in our case studies in Fig. 3. The Clip mechanism mitigates the intrinsic biases of the PRM. The Delta mechanism tackles the reward misspecification issue. **As these mechanisms focus on tackling different aspects, they can be readily combined to enhance RL training.**
>
>
> ## 3. Performance Gain of the Proposed Mechanisms
> > "Can we consider the gains by mixing the clip and delta mechanism are strong and not modest?"
>
> - Yes, the gains by mixing the Clip and the Delta mechanisms are strong.
> - Our ablation study in Table. 1 and training curves of all methods in Appendix A.2 (also [available here](https://i.postimg.cc/jd1HPLQM/training-curves.jpg)) shows that PR-Clip-Delta can achieve higher greedy accuracy and sampling accuracy than the baseline methods.
> - The effectiveness of the proposed approaches can be better illustrated in Fig. 5 (or  [available here](https://i.postimg.cc/qRTLfCDT/perf-improve.jpg)), where **using PR-Clip-Delta as dense rewards improves RL training with success rewards only across all evaluated LLMs.**
>
> ## 4. Discussion on the Hackability of PRM
>
> > "Your PRM is trained on automatic data. Is this possible that this hackability issue is because of the noise caused by the automatic procedure and would be avoided when trained on human-data or better PRM data generation (I understand better PRM data generation is itself a research question)?"
>
> - We believe that training on human data or better PRM data can mitigate but not completely solve the identified issues. Intrinsic bias generally exists for learned reward models, not just limited to PRM. On the other hand, though the training data has noise caused by the automatic procedure, human data itself also has noise due to the diverse preferences of the labelers. In practice, we also find that a public PRM trained on human data (specifically, llemma-7b-prm-prm800k-level-1to3-hf) can also have intrinsic bias, assigning non-negligible and even high values to incorrect steps, as shown in Appendix F. We also believe that, if a better PRM is available, our approach can be further applied to enhance the performance of stronger LLMs through RL training.
>
> We hope our response addresses your concerns, and we welcome any further questions or suggestions.

---

> > ### Comment · Reviewer_7bHH · 2024-11-23
> >
> > Thank you for your rebuttal.
> >
> > I still find the delta mechanism unmotivated. Specifically, I don't understand this sentence "The Delta mechanism ensures the steps promoted by RL training are aligned with the PRM, which promotes the correct step in this case." The delta mechanism changes the rewards in such a way that the sum of the altered (after delta) rewards, .i.e.  the altered return, becomes equal to the last reward essentially (right?). I still don't get why that should be useful.
> >
> > The clipping mechanism makes sense as you just don't want the PRM to offer much rewards as it happens to be not really trustworthy.
> >
> > However, still the mixing of the two becomes confusing. From the training curves, I can see that it is only the clip+delta that achieves a higher test accuracy, which is the only accuracy that really matters (right?). Therefore, I don't see the delta or the clip improving the test accuracy alone. I am looking at epoch 5 because I don't know why we should look mid-training and not the final outcome.
> >
> > I thank the authors. I think pointing the issue of PRMs is important and that is the main reason I gave the paper a 6. Unfortunately, I don't find the paper motivated enough to give an 8. If anything, a 6 is a bit on the upper side as well as I don't find the proposed solution to be generalizable or motivated and at the end the improvements are marginal.

---

> > > ### Author Response · Authors · 2024-11-23
> > > **Further Clarification**
> > >
> > > We really appreciate the reviewer's immediate feedback. We would like to explain the motivation of the Delta mechanism further.
> > >
> > > ## Return of the Delta Mechanism
> > >
> > > > "The delta mechanism changes the rewards in such a way that the sum of the altered (after delta) rewards, .i.e. the altered return, becomes equal to the last reward essentially (right?). I still don't get why that should be useful."
> > >
> > > We emphasize that, for any intermediate reasoning step $k$, **the Delta mechanism redefines the return of the altered reward to be equivalent to the PRM reward at step $k$, rather than at the last step**.
> > >
> > > This can be better illustrated by analyzing the return of the Delta mechanism, for any $1\le k\le K-1$
> > > $$
> > > \text{Return}(q,p^{(k)})=\left[\sum_{k'=k}^{K-2}r_{process}(q,p^{k'})-r_{process}(q,p^{(k'+1)})\right]+r_{process}(q,p^{(K-1)})=\underbrace{r_{process}(q,p^{k})}_\text{PRM reward at step k instead of the last step}
> > > $$
> > >
> > > Therefore **RL training would optimize the single-step PRM reward for any intermediate step** rather than at the last step. Since the PRM is trained to predict the quality of one step, optimizing the PRM reward could enhance each reasoning step instead of the last step only.
> > >
> > > Another interpretation is that, as we prove in Appendix F.2, PRM (trained with automatically generated labels) are actually learning the value of a stronger policy. With the Delta mechanism, for any intermediate step, RL training optimizes the probability of correctly completing a solution by using a stronger policy.
> > >
> > > ## Mixing the Clip and the Delta Mechanisms
> > >
> > > Since the Clip and the Delta mechanisms focus on tackling issues from different aspects, as we analyzed in Sec 4, each of them alone can not tackle all issues. In practice, we can observe the Clip and the Delta alone can improve the training, but the optimization process could be less stable, as shown in Fig. 10. Combining both approaches can tackle the issues we identified and achieve a stable improvement on the RL training, as shown by the curve of PR-Clip-Detla in Fig. 10.

---

> > > > ### Comment · Reviewer_7bHH · 2024-12-01
> > > > **Response to Rebuttal**
> > > >
> > > > Dear Authors,
> > > >
> > > > Thank you for the explanation on the delta mechanism. Here is my understanding: The PRM gives rewards for each step. The delta mechanism acts on top of the PRM rewards? So, it changes the rewards for the step to "the change in the rewards of a step". In your setting, if the solution is step1, step2, step3 and the PRM rewards are r1, r2, and r3, you are changing it to 0, r2-r1, r3-r2. That is what I am saying is unmotivated. The PRM itself should describe whether a step is good quality. But, you change it to whether the step was higher quality than the previous step.
> > > >
> > > > As for my score, I will keep my score the same because I think 6 is on the higher end and the reason for the 6 is the importance of pointing out PRM shortcomings.

---

> ### Author Response · Authors · 2024-12-01
>
> Dear Reviewer,
>
> Thank you for your feedback and for your understanding of the delta mechanism.
>
> We emphasize that **an RL algorithm** promotes steps with **high returns** instead of **high step-wise rewards**. Here the return refers to the rewards accumulated over the course of the solution.
>
> The Delta mechanism indeed operates the rewards. It calibrates how the RL algorithm processes the rewards. In our example shown in Fig. 3, although a correct step has a high PRM reward, its return is much lower due to a shorter solution. Therefore, the RL algorithm might erroneously favor the incorrect step with a low PRM reward. With the Delta mechanism, we correct the **returns** so that they are **consistent with the PRM rewards**.
>
> Please refer to Fig. 3 and Appendix E.1 in the revised paper for more details.
>
> We hope this explanation clarifies the motivation behind the delta mechanism.

---

### Author Response · Authors · 2024-11-21
**Global Response**

We sincerely thank all reviewers for their insightful and constructive feedback. In response, we have substantially revised and enhanced our paper to address the concerns raised and to further strengthen our contributions. Below, we outline the key updates and clarifications:

## 1. New Case Studies and Theoretical Analysis
We highlight key aspects of our revision as follows:
- **New Case Studies on Issues of PR (Fig. 3)**: We provide new in-depth case studies in Fig. 3 (also [available here](https://i.postimg.cc/Nj894gSL/case-study.jpg)) showing issues when the PRM simply serves as reward shaping in addition to the success reward. Specifically:
    -  **Intrinsic biases of PRM** can be largely exploited by the training LLM to generate sub-optimal behavior patterns.
    - **Reward Misspecification Issue** could mistakenly promote incorrect steps through RL training.
- **Additional Mechanism Analysis (Sec 4.2)**: Our detailed analysis of the mechanisms highlights how the proposed mechanisms address specific issues,
    - **Clip mechanism mitigates the intrinsic biases of PRM** by bounding the rewards to an upper threshold, preventing the LLM from obtaining high rewards through undesired patterns.
    - **The Delta mechanism tackles the reward misspecification issue** by optimizing single-step PRM rewards, ensuring the steps promoted by RL training are aligned with the PRM.
- **Theoretical Insights for the Delta Mechanism (Appendix E)**: We provide theoretical analysis showing how the Delta mechanism makes RL training optimizates single-step PRM rewards alongside success rewards.
- **Additional Experiments & Training Curves (Appendix A)**: Additional experiments and training curves help better understand the limitations of baseline methods.

## 2. Novelty of Our Work
We highlight the **novelty of our work** as follows:
- **The First Study of PRM Issues in RL Training for LLM Reasoning:** Our work presents the first thorough study of the issues when simply adopting PRM as reward shaping in RL training for LLM reasoning. We find the intrinsice biases of PRM and the reward misspecification issue may lead to sub-optimal performance. The impact of these issues on RL training for LLM reasoning tasks has not been systematically analyzed in prior literature [1,2,3,4].
- **Simple Techniques and Practical Insights for Effective RL Training:** Although the proposed mechanisms are simple in design, they are supported by thorough analysis and experiments that show their ability to utilize PRM effectively, providing valuable rewards in RL training for LLM reasoning. Our studies and analysis present valuable insights to the community on how to design effective RL rewards for LLM reasoning.

## 3. Summary of Major Revisions
- **Fig.3**: Updated with new case studies and insights into mechanism effects.
- **Sec 4.1**: Discussion on the training results of PR \& OR. Additional case studies on the issues of PR.
- **Sec 4.2**: The motivation and detailed explanation of the Clip and the Delta mechanism.
- **Sec 5**: Updated PPO training results with success rewards only in Table 2.
- **Appendix A**: Training curve and additional experiments.
- **Appendix D**: Hyperparameters and training setup.
- **Appendix E**: Theoretical analysis of the Delta mechanism and PRM.


[1] Shao, Zhihong, et al. "Deepseekmath: Pushing the limits of mathematical reasoning in open language models." arXiv preprint arXiv:2402.03300 (2024).

[2] Yang, An, et al. "Qwen2. 5-math technical report: Toward mathematical expert model via self-improvement." arXiv preprint arXiv:2409.12122 (2024).

[3] Wang, Peiyi, et al. "Math-shepherd: Verify and reinforce llms step-by-step without human annotations." arXiv preprint arXiv:2312.08935 (2023).

[4] Havrilla, Alex, et al. "Teaching large language models to reason with reinforcement learning." ICML 2024 Workshop AI4MATH.

---

### Author Response · Authors · 2024-11-25
**Motivation and Novelty of the Delta Mechanism**

To address the common questions regarding the Delta mechanism, we would like to clarify the motivation behind the Delta mechanism and its core novelty.

The key novelty of the Delta mechanism lies in its ability to **guide RL training to optimize the single-step PRM reward for each individual reasoning step**. By contrast, PR encourages longer generations, even when the PRM rewards of individual steps are sub-optimal, as shown in our case studies (Fig. 2 and Fig. 3).

The contrast becomes clearer through an analysis of the policy gradient for PR and PR-Delta. Following [1], the policy gradient of RL training can be given in a step-wise manner,
    $$
        \nabla_\theta J_{r}(\pi_\theta)=\mathbb E_{q\sim \mathcal D,s\sim \pi_\theta(\cdot|q)}\large[\nabla_\theta \log\pi_\theta(s|q)\cdot \text{Correct}(q,s)+ \alpha\cdot\sum_{k=1}^{K}\nabla_\theta \log\pi_\theta(s^{(k)}|q,p^{(k-1)})\cdot \text{Return}(q,p^{(k)})\large] +\text{KL term}
    $$
where $\text{Return}(q,p^{(k)})$ denotes the return starting from step $k$.


1. **RL Training with SR+PR.**
    The return is given by,
    $$
        \text{Return}(q,p^{(k)})=\underbrace{\sum_{k'\ge k}r_{process}(q, p^{(k')})}_{\text{Sum of PRM rewards}}
    $$

    This formulation encourages producing a larger number of reasoning steps, even when the PRM rewards are low. For example, an incorrect solution with 10 steps and an average PRM reward of 0.3 could be preferred over a partially correct solution with only 3 steps and a higher average PRM reward of 0.8. The RL training thus prioritizes the longer solution with lower average rewards.

2. **RL Training with SR+PR-Delta.**
    Introducing the Delta mechanism, the return of PR-Delta starting from step $k$ is adjusted by,
    $$
    \text{Return}(q,p^{(k)})=\left[\sum_{k'=k}^{K-2}r_{process}(q,p^{k'})-r_{process}(q,p^{(k'+1)})\right]+r_{process}(q,p^{(K-1)})=\underbrace{r_{process}(q,p^{(k)})}_\text{PRM reward at step k}
    $$
    Here, the training optimizes **the PRM reward for each intermediate step** rather than focusing on the aggregated reward. Since the PRM is trained to predict the quality of one step, this approach could enhance the reasoning process step by step.

**For further details, please refer to the revised paper, particularly Section 4.**

[1] Sutton, Richard S., and Andrew G. Barto. Reinforcement learning: An introduction. MIT press, 2018.

---

### Meta-Review · Area_Chair_fbvy · 2024-12-23

**Metareview:**

This paper presents a way to utilize process reward models (PRMs) for RL training in LLM reasoning. The paper studies some issues with PRM rewards, followed by a discussion of how adjusting process rewards via clipping and delta mechanisms can help improve performance. The paper presents interesting takeaways -- including positive biases of the PRM, reward hacking of the PRM, and shows performance with their suggested changes to PRM rewards (though some ablations are still not clear; see discussion with Reviewer 7bHH).

While I enjoyed reading the paper, there're some weaknesses: (1) it is unclear if these findings hold true for any PRM, (2) in my opinion and in the opinion of some reviewers as well, the analysis section does not decouple issues with using process rewards vs errors in the process reward model itself, (3) connections to RL literature are not made explicit, and (4) the reviewers are not totally convinced by the ablations for the method proposed (although to the authors' credit, the proposed method is simple).

Unfortunately while the paper has the potential to be quite impactful, due to the above issues we are not able to accept the paper at this moment.

**Additional Comments On Reviewer Discussion:**

The reviewers raised some valid points regarding deeply understanding the issues with PRMs and the efficacy of the approaches proposed by the authors. I believe that a lot of the points raised are valid, and while the authors did address several of these points, as I mention above, there are some points which are still not made explicitly clear.

The only reviewer to champion the paper provides a low confidence of 2, and I do agree with the points raised by other reviewers.

---

### Decision · Program_Chairs · 2025-01-22

Reject